# Trackable and scalable LC-MS metabolomics data processing using asari

Shuzhao Li [1,2] ✉, Amnah Siddiqa[1], Maheshwor Thapa[1], Yuanye Chi[1] & Shujian Zheng[1]

Significant challenges remain in the computational processing of data from liquid chomratography-mass spectrometry (LC-MS)-based metabolomic experiments into metabolite features. In this study, we examine the issues of provenance and reproducibility using the current software tools. Inconsistency among the tools examined is attributed to the deficiencies of mass alignment and controls of feature quality. To address these issues, we develop the open-source software tool asari for LC-MS metabolomics data processing. Asari is designed with a set of specific algorithmic framework and data structures, and all steps are explicitly trackable. Asari compares favorably to other tools in feature detection and quantification. It offers substantial improvement in computational performance over current tools, and it is highly scalable.

Metabolomics holds the promise to comprehensively measure and quantify small molecules in biological systems. Since the chemistry of these small molecules underlies most aspects of life science, metabolomics is recognized as critical to support missions in biomedical research, including precision medicine and environmental health[1–3]. Since the mid-2000s, the experimental platforms of metabolomics have improved significantly, and LC-MS (liquid chromatography coupled mass spectrometry) has become the leading technology.

In LC-MS metabolomics, a sample is scanned by a mass spectrometer consecutively during the chromatography, generating a time series of spectra, each containing a list of ions with mass to charge ratio (m/z) and intensity values. The goal of data processing is to report a quantitative value per metabolite feature per sample, which is a proxy for its biological concentration. Multiple software tools have been developed for LC-MS metabolomics data processing over the years[4–11], and the most widely used are XCMS[9] and MZmine[7]. They have contributed significantly to the growth of the field, but major design issues have also become apparent over the past decade.

These software issues are manifested in the current roadblock of reproducibility, which has greatly limited the adoption of metabolomics technologies by the broader research community. In global profiling by a typical high-resolution mass spectrometer, studies report 1000 s to 10,000 s of features. Myers et al.[12] compared XCMS and MZmine 2, and found that half or more of the features were not shared between the tools. Delabriere et al.[13] reported similar level of

disagreement between XCMS and OpenMS. It is common knowledge among users that the results also vary wildly based on parameter settings. Significant community efforts were spent on parameter optimization of XCMS[13–18]. However, these post-hoc adjustments do not address the fundamental design issues, namely, strong dependence on complex parameters, feature correspondence errors in larger studies and difficulty in tracing software issues.

Additionally, as a key−omics technology, the users of metabolomics cannot be limited to chemists, rather, the data processing must be trusted by general bioinformatics practitioners. This requires a set of transparent quality metrics and traceability through every major step of data processing. These requirements mirror principles from modern software design where explicit linking between steps is necessary for automated testing, debugging and continued improvement. Robust design combined with transparent algorithms will certainly help address the roadblock of reproducibility in metabolomic studies.

In this study, we present asari, an open-source software tool for LC-MS metabolomics data processing. Asari is designed with a set of distinct algorithmic framework and data structures, and all steps are explicitly trackable. To take advantage of the mass resolution in current data, mass alignment is performed first, represented by mass tracks within and composite mass tracks across acquisitions in an LC-MS metabolomics experiment. They facilitate, along with a set of quality metrics, the better understanding of reproducibility in detecting mass peaks and elution peaks, and correspondence of LC-MS

[1]Jackson Laboratory for Genomic Medicine, 10 Discovery Drive, Farmington, CT 06032, USA. [2]University of Connecticut School of Medicine, Farmington, CT, USA. ✉e-mail: shuzhao.li@jax.org

features. Asari offers substantial improvement of computational performance over the previous tools, and is highly scalable.

## Results

### Provenance issues in feature correspondence during LC-MS data preprocessing

Metabolomics today usually employs high-resolution mass spectrometers that are often capable of mass resolution at 5 ppm (part per million) or better. This means that the measurement error for a singly charged molecule of 150 Dalton is no greater than 0.00075 in m/z values (150 * 5 * 1E-6), or 0.0040 m/z for a molecule of 800 Dalton (800 * 5 * 1E-6). Previously, mass spectrometry software used to collapse data into m/z bins of nominal or 0.1 amu (atomic mass unit). With the mass resolution in today's data, binning is no longer a valid approach, and the mass should be reported as precisely as possible to support compound identification. This step is the detection of mass peaks: a group of ions measured on the same molecular species have small random variations, and a consensus m/z value is determined for the group as a mass peak. The detection of mass peaks now is usually performed by the "centroiding" process; centroided data are then used as input to metabolomics data processing. Centroiding is supported by all major instrument manufacturers. Tools like ThermoRawFileParser[19] and msConvert[20] are commonly used both for data format conversion and for centroiding.

A feature in LC-MS metabolomics is defined by a unique pair of m/z value and retention time in chromatography. The m/z value of a peak is first determined per sample. When peaks are later aligned cross samples (i.e., correspondence), the m/z values vary slightly in each sample and the algorithm must ensure the correct peaks are grouped and report a consensus m/z value. There are more steps in data processing, while we first focus on m/z alignment.

We generated an LC-MS metabolomic dataset (HZV029) of 184 repeated analyses of a pooled human plasma sample. The same data were processed by XCMS and MZmine, using similar parameters. The two tools produced very different numbers of features, and about 60% of XCMS features are unambiguously matched to MZmine features (Fig. 1a). Similar levels of disagreement are seen in MZmine using a different algorithm and MS-DIAL (Supplementary Table 1), and in a different dataset (Supplementary Table 2).

Different from earlier studies, almost all XCMS feature here can find a counterpart in the MZmine result, within 5 ppm of m/z and 6 s of retention time. This is because HZV029 has better data quality than older studies, and it's significantly larger. If a feature is missed in 1 sample, it is unlikely for the processing software to miss it in the other 183 samples. The problem here is that many features are not uniquely matched. Here, a unique match is defined as a pair of features that are reciprocally best matches in the allowed m/z and retention time windows. An ambiguous match is shown in Fig. 1b, where 3 XCMS features are indistinguishable from 5 MZmine features. The possible reasons for

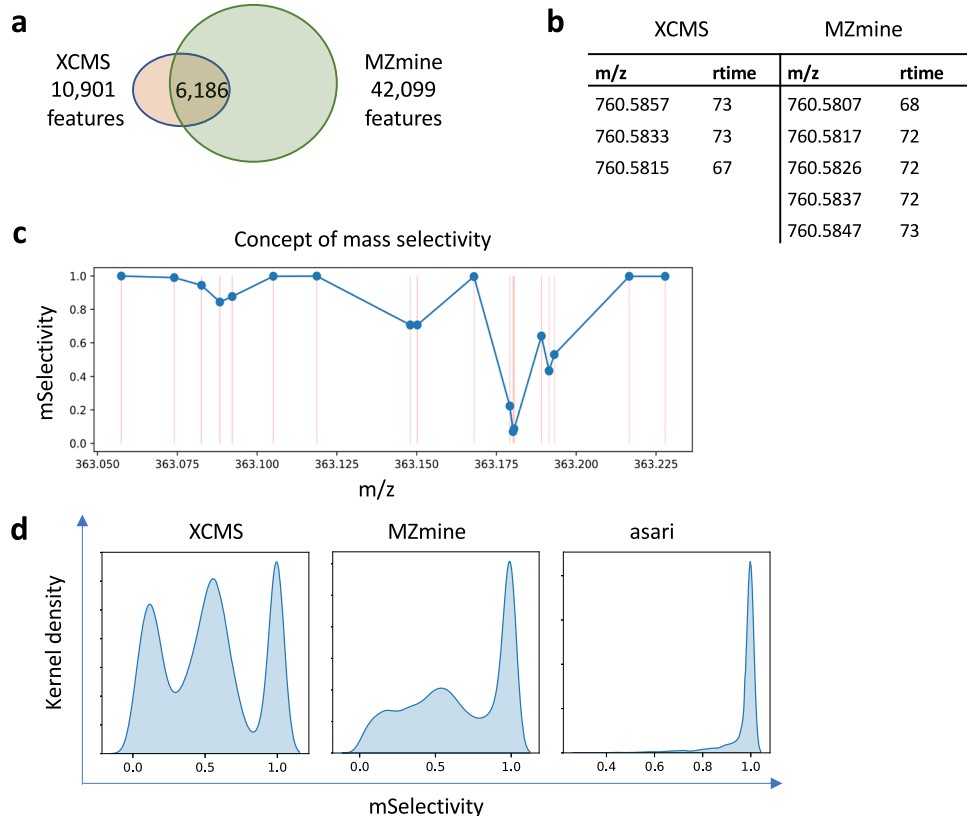

**Fig. 1 | Provenance issues of feature correspondence in LC-MS metabolomics data processing. a** On a dataset of 184 repeated samples analyzed on an Orbitrap ID-X mass spectrometer (HZV029), XCMS generated 10,901 features and MZmine generated 42,099 features (see Supplementary Methods for versions and parameters). Between the two results, 6186 features are uniquely matched. Additional comparisons are reported in Supplementary Tables 1, 2. **b** Many mismatched features are due to failure to resolve reciprocal best match. This example shows all 3 features in XCMS match to all 5 features in MZmine. Retention time (rtime) is in seconds. **c** Illustration of mSelectivity as a function of how distinct a m/z value is, regarding to its neighboring features and mass resolution. Each dot represents a m/z feature, and its mSelectivity value (Y axis) depends on the horizontal distance to neighbor features. The error in matching m/z values is modeled as a Gaussian distribution dependent on mass resolution, and mSelectivity is low when a feature has neighbors with close m/z values. **d** Distribution of mSelectivity in the features produced by three processing tools. Feature m/z values are rounded to the 3rd decimal place, so that minor variations are ignored and split peaks are not taken into account. Rounding has no impact on asari, because asari m/z values are linked to mass tracks. Source data are provided as a Source Data file.

this type of mismatch include (a) split mass peaks on the same feature, (b) failure in m/z alignment cross samples, and (c) that multiple features do exist because of close chemical properties. The last scenario does occur but not in high frequency. All three scenarios violate the mass resolution of 5 ppm set for this experiment, i.e., 760.5807 and 760.5817 cannot be distinguished on the mass spectrometer. If the processing software does not resolve the issue, i.e., reporting a unique numerical value for the same mass, errors propagate into downstream annotation and statistical analysis. The confusion in comparison between tools reflects their internal confusion in feature correspondence, more specifically, mass alignment.

Using a similar concept to selectivity in mass spectrometry, we define a "mSelectivity" function to calculate how well a m/z feature is distinguished from others under a given mass resolution (see Supplementary Methods). The concept of mSelectivity is illustrated in Fig. 1c, where a m/z feature has a low mSelectivity score when it has neighbors of very close m/z values. We computed mSelectivity scores for the features from XCMS and MZmine, and their distributions are shown in Fig. 1d. These mSelectivity scores were computed after their m/z values were rounded to the 3rd decimal place and collapsed into unique lists, to forgive rounding errors and exclude possible isomers (i.e., compounds of the same mass). Figure 1d indicates that a very large number of features in XCMS and MZmine have poor mSelectivity that is not consistent with the resolution of instruments. The ideal result is that all features have mSelectivity close to 1, which is perfect compliance of mass resolution. That is how we implemented mSelectivity requirement in asari (right panel in Fig. 1d).

The distribution of mSelectivity is a straightforward summary of how well software tools distinguish m/z values in the metabolomics feature tables. The poor characteristics of XCMS and MZmine in Fig. 1d reflect artifacts in feature correspondence, because the mass peaks detected in a single sample prior to correspondence do not have as severe a problem. Retention time in chromatography may help distinguish compounds of similar m/z values, but it does not fix issues in m/z alignment in the software, and the sheer number of problematic m/z values causes significant problems in reproducibility.

In practice, an ad hoc step could be performed to merge these close features. XCMS has a step of merging neighbor peaks that are too close, which is among the reasons of fewer features reported here. But Fig. 1d shows clearly that it does not solve the real problem of mass selectivity. Furthermore, ad hoc merging is a subjective approach, and usually reserved for expert users. It causes issues in the quantification of these features implicitly because merging alters processing history and it does not affect all peaks equally. Most critically, ad hoc merging does not help the reproducibility of software.

## Mass alignment should not be conditioned on elution peak detection in high resolution metabolomics

The design of XCMS and MZmine is similar: extracted ion chromatograms (EICs or XICs) are built on regions of interest (ROIs) in each sample; elution peaks are identified on each EIC; and elution peaks are aligned across samples to become features, i.e., feature correspondence (Fig. 2a, left). Here, m/z alignment occurs after detection of elution peaks in each sample. In this approach, two peaks in a sample can start with the same m/z value but end up with different m/z values after correspondence. We note that the construction of chromatograms is straight forward in high resolution data, but detection of elution peaks is very error prone (discussed in a later section). The problem may not be pronounced in low-resolution data with few samples, but is amplified exponentially by a large number of peaks and a large number of samples. Without tracking EICs explicitly, it is a poor design to perform m/z alignment after elution peak detection.

We implemented the "mass track" concept to address this issue. A mass track is defined as a series of LC-MS data points of the same consensus m/z value and spanning the full retention time (Supplementary Fig. 1). A mass track may or may not have detectable elution peaks, and zeros are filled for scans without an intensity value. Therefore, a mass track is an EIC that spans the full retention time, serving as the parent level of peaks. Mass tracks are aligned cross samples before elution peak detection in asari (Fig. 2a, right). This takes advantage of the superb mass resolution on today's instruments and avoids errors stemming from elution peak detection. Mass tracks lock in unique m/z values per sample, therefore greatly reducing the complexity of m/z alignment, compared to the alignment of individual elution peaks. The mass tracks that match 13 C/12 C and Na/H patterns are considered of high confidence and used as guides in the m/z alignment.

## Mass tracks and composite mass tracks anchor LC-MS alignments in asari

In biological studies, a feature is expected to have variations and it is common that it is under detection limit in many samples. Even if no elution peak is detectable, mass tracks still often exist in those samples, and the information is useful towards the correct m/z alignment. On the other hand, feature detection should utilize the recurrent pattern of the same peak in many samples. In asari, aligned mass tracks are summed into a "composite mass track" after correction of retention time (Fig. 2b). With this approach, elution peak detection is no longer required on individual samples. It can be done on the composite mass track, then the peak area is looked up in each individual sample and reported as feature intensity values (Fig. 2b).

This greatly simplifies feature correspondence, and has a significant performance gain, by not repeating the computational cost of peak detection on all individual samples. Because the composite mass track has higher signals than any individual sample, the quality of peak detection is often improved. In practical terms, detecting weak peaks will be enhanced by combining signals from multiple samples; irregularity such as chromatographic gap and bad peak shape will be ameliorated (example in Supplementary Fig. 2).

The overall design of asari is illustrated in Fig. 2c, with a detailed description in the Supplementary Information. Aligned mass tracks of all samples form a "MassGrid", which provides a foundation for trackable and high-quality feature correspondence. Retention time (RT) correction between samples, also called retention time alignment, is carried out by LOWESS regression, using a small subset of high-quality elution peaks. These reference peaks are easy to select from previously aligned mass tracks, preferably as the only peak on a mass track. The regression result is translated to an RT remapping function for each sample. This remapping is bidirectional: RT on the composite mass track is mapped back to each sample correctly regardless how much error is in the regression model. The error only affects the peak range on the composite mass tracks, which is mostly inconsequential. RT alignment has no impact on m/z alignment.

In previous tools, if a feature is only present in one or few samples, correspondence becomes problematic. However, the presence of a single good elution peak is evidence that the analytical method is valid for that particular metabolite. It is important to know that weak or no peaks in other samples is reflecting biology not inadequate chemical analysis. When many samples do not have a good signal on a feature, it poses a significant challenge in peak detection individually. Since asari considers all mass tracks in the composite map as combined signals, it largely bypasses this challenge. Even if a peak is present only in a single sample, it will be detected and reported in asari, which is important to applications such as personalized medicine and exposomics.

## Composite map enables easy and interactive inspection of data

The design of asari significantly improves provenance through data processing. To track a problem in software, computational objects in each step must be explicitly linked while each step shall be verified separately. Currently, it is cumbersome to verify peaks from XCMS. It usually requires scripting to plot each ROI associated with a feature,

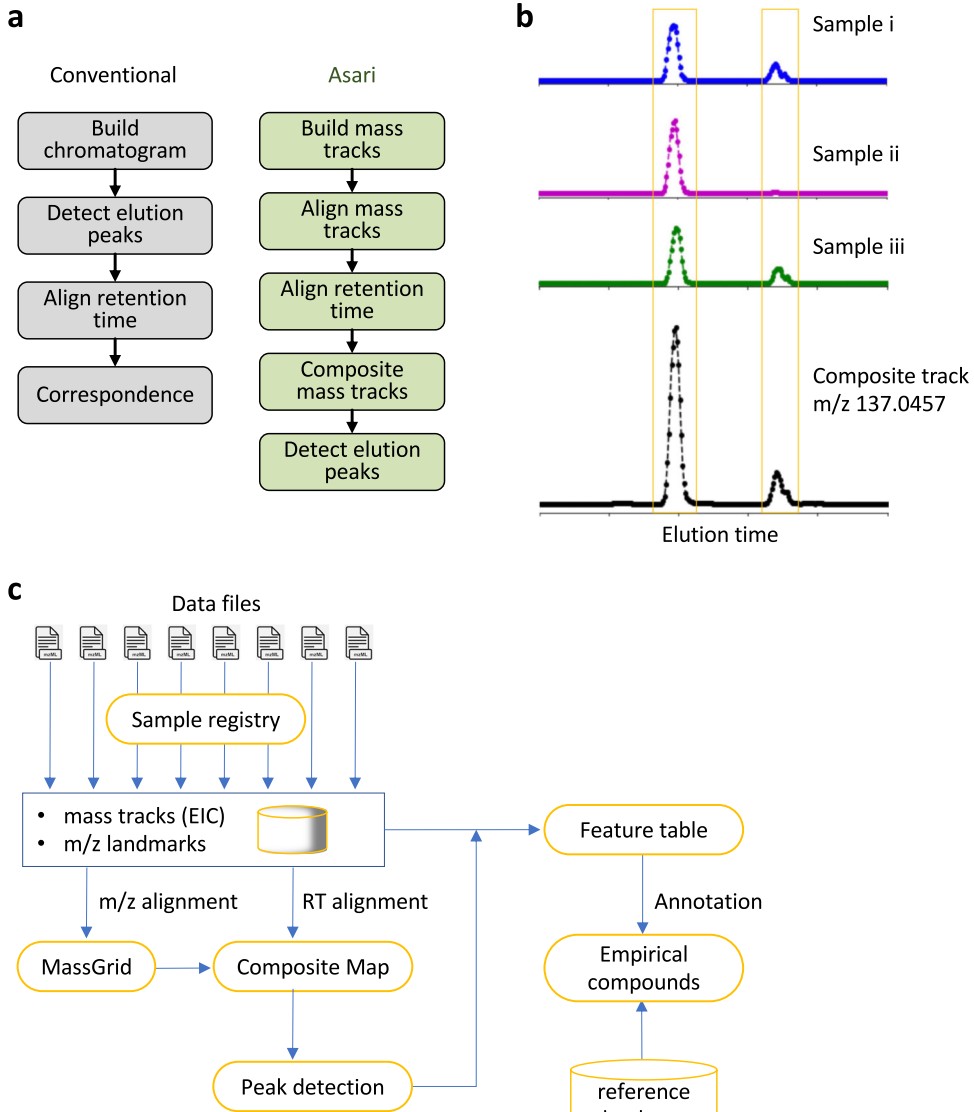

**Fig. 2 | The design of asari anchors on composite mass tracks. a** Overview of design differences between extant software tools and asari. **b** The "composite mass track" is a representation of data from all samples, by adding up the signals in corresponding mass tracks after retention time (RT) alignment. Mass tracks from the MT02 dataset are used as an example. **c** Asari takes centroid mzML files as input and builds chromatograms for each as mass tracks. To prioritize modern mass resolution, m/z alignment is performed first to form a MassGrid, aided by isotopic landmarks. Aligned mass tracks across samples are corrected for RT, using LOWESS regression on a subset of high-quality elution peaks, then aggregated into composite mass tracks (**b**). All composite mass tracks are stored in the "Composite Map". Elution peak detection is performed on the composite mass tracks, and feature table is generated by looking up the corresponding peak areas in each individual sample. Annotation groups degenerate features into empirical compounds, and reference databases are used to match the m/z values in empirical compounds.

which is not real backtracking because an ROI is not the same as EIC and has more data points. On the other hand, the demand of computational resources increases quickly if all intermediate steps are recorded (part of the performance penalty in MZmine). By linking features and raw data through composite map, asari produces an efficient solution for automated testing, debugging and verification. A significant change is that asari composite map is representative of all samples – the burden of peak evaluation and visualization is largely removed from repeats on individual samples. Therefore, this enables a data dashboard for users to navigate and inspect data, regardless of the size of samples.

An example asari dashboard is shown in Fig. 3a. This is an interactive tool that users can launch into a web browser after each dataset is processed. The top part of the dashboard is a set of tabs for data summary and quality metrics. Users can navigate through the feature browser to inspect EICs by clicking, hovering, panning and zooming

functions. A separate mass track viewer shows all detected peaks on a mass track, with an example screen shot shown in Fig. 3b. When users zoom in different regions of the mass track, the peaks can be inspected closely (Fig. 3c, d). The ability to inspect data and feature quality visually is important for the reproducibility of science, and should be done before resources are committed to continued work. The asari dashboard is not only useful to end users, but also to developers and data scientists for testing and interacting with the software.

**Elution peak detection is verifiable and understandable in asari**
The design of asari leads to the improvement of multiple key aspects. We start with benchmarking feature detection, before discussing the underlying mechanistic of elution peak detection and quality metrics. Of note, the approach here on feature detection does not concern false positives, which are discussed in the next section. The benchmark datasets must be impartial to the tools being tested.

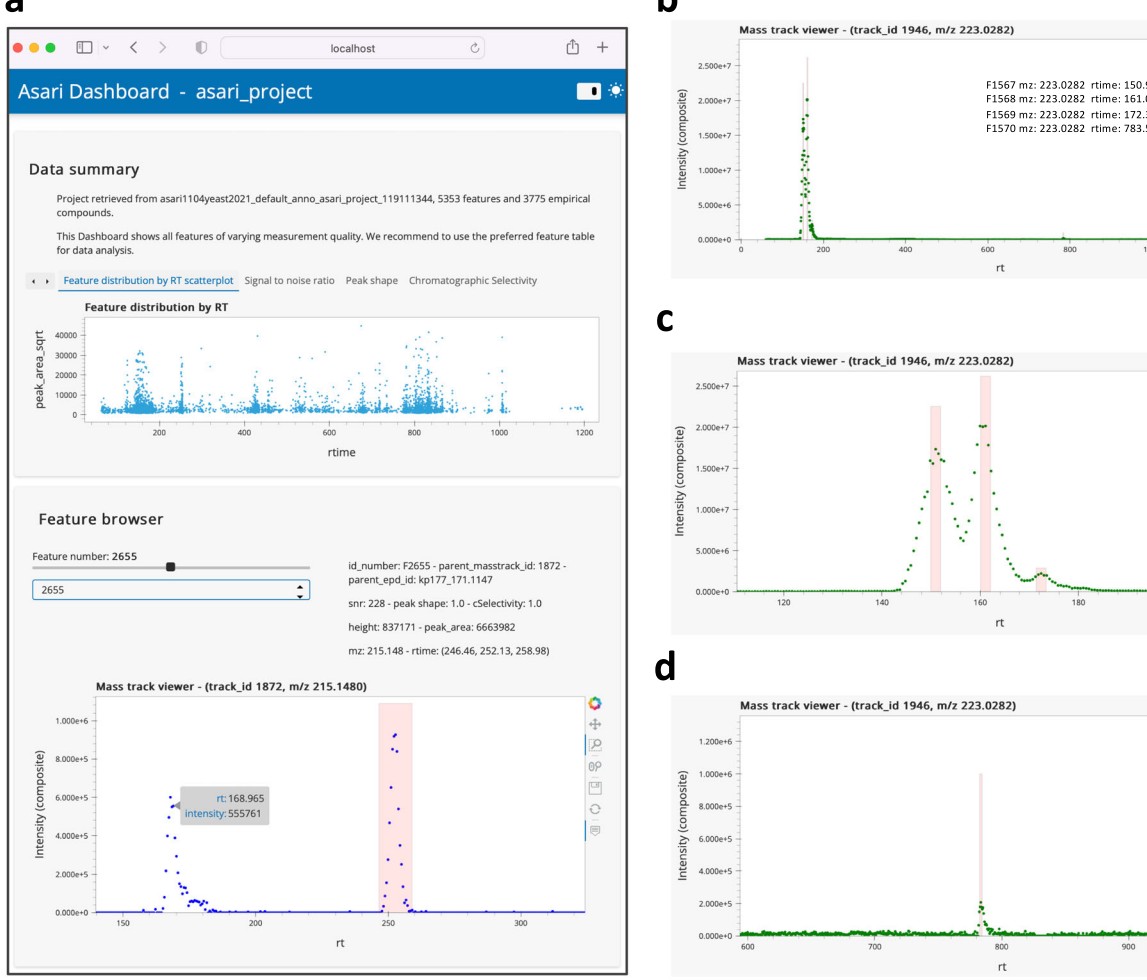

**Fig. 3 | Composite map enables easy and interactive inspection of data in a web browser.** Users can click, zoom, pan and hover on the interactive figures. Raw data points are plotted without smoothing. The dashboard can be launched by the asari viz subcommand. **a** Screen shot of the asari Dashboard, with the Feature browser. Users can click through all features or find by feature ID. **b** A mass track with four peaks and zoom-ins are shown in **c** and **d**. Mass track viewer supports m/z search.

We used the manufacturer's software to process, select and visually verify 402 features in the HZV029 data. The detection of these features by each tool is reported in Fig. 4a. In its default setting, asari detected 386 of these features, the second highest next to MS-DIAL. Of the 16 features asari disagreed with the manual list, 7 were on low quality EICs discarded by asari, and 9 were peaks of poor shape and borderline number of data points (Supplementary Fig. 3). We note that the higher number of detected features by MS-DIAL is attributable to its low stringency by reporting 29,924 features, several times more than others. All the 8 peaks missed by MS-DIAL, however, had good peak shape and were detected by asari (Supplementary Fig. 4).

A second benchmark dataset is Yeast2021, in which Chen et al.[21] manually confirmed 314 identified features. The detection result by each tool is shown in Fig. 4b. Asari returned the best performance of detecting 310 of the 314 features. The 4 features missed by asari are shown in Supplementary Fig. 5: two of them have too few data points, one under minimal requirement of peak height, and one with too high local noise. The $^{13}C$ isotopologues and $Na^+$ adducts associated with the known compounds are shown in Supplementary Fig. 6.

Additional comparisons were performed between XCMS and asari. We have analyzed the human plasma reference sample NIST SRM 1950, and verified 39 metabolites previously reported in the literature[22]. Both asari and XCMS successfully detected all these 39 metabolite features (see Supplementary Methods). We generated a new dataset based on credentialed *E. coli* samples (Fig. 4c, similar to

ref. 23). A subset of E. coli metabolites was labeled by $^{13}C$ isotope during the cell culture. XCMS and asari extracted 1525 and 1399 features from this dataset, respectively. After matching isotopic patterns between labeled and unlabeled samples, 643 features in total showed correct isotopic patterns as protonated ions. Among them, XCMS detected 581 and asari 621. Among the 22 features missed by asari, two features were out of the 5 ppm m/z range; the other 20 are plotted in Supplementary Fig. 7. Three of them would pass as real peaks with more aggressive smoothing, and the remaining 17 features do not meet quality requirements in asari. Taken together, these results indicate that the peak detection performance of asari compares favorably against others, and that the behavior of asari is understandable and predictable.

**Low quality peaks account for major discrepancy between different software tools**

Elution peak detection deserves a detailed investigation, because there are many confusions in the field and the coverage of metabolomics is related to how many peaks are detected. As seen in Fig. 1 and Supplementary Tables 1, 2, the numbers of features are quite different from different tools, even though XCMS and MZmine have the same wavelet algorithm[24]. Therefore, the differences are not solely caused by peak detection algorithms. Above, we discussed the impacts from mass alignment and chromatogram construction. When different peak detection algorithms are used within MZmine, using the same EICs, they

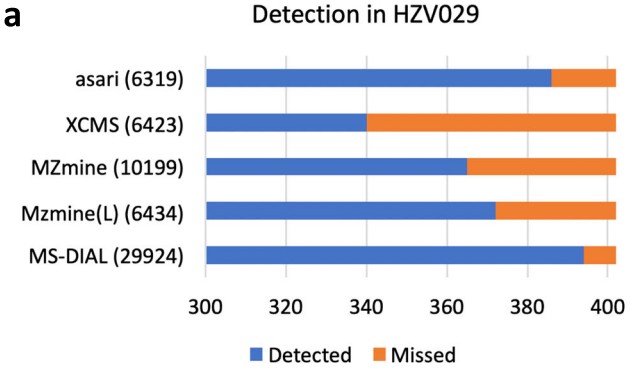

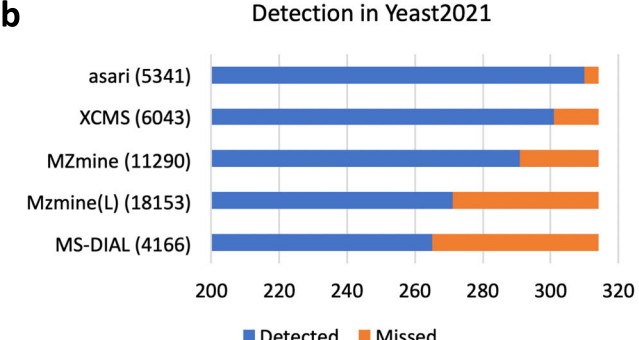

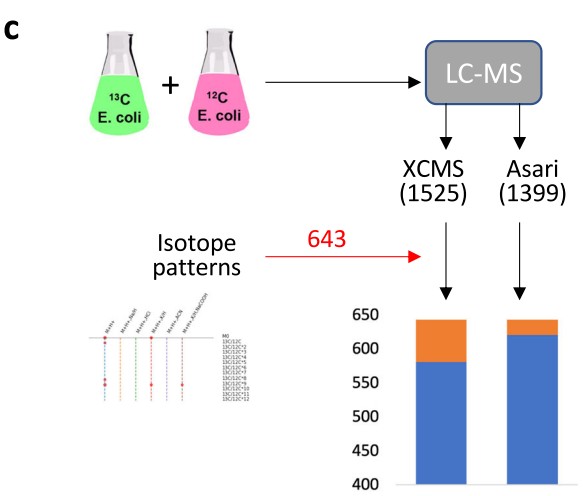

**Fig. 4 | Evaluation of asari feature detection. a** Detection in HZV029 dataset, based on three randomly selected samples. The manually verified 402 features were selected from results by Thermo Scientific software. **b** Detection of manually certified features in three samples in the Yeast2021 dataset[21]. MZmine version 3.3.0 (wavelets) produced identical results as version 2.53 in both datasets (A and B). **c** Detection in the credentialed *E. coli* data. XCMS and asari used similar parameters (peak height > 1E5). The two feature tables were annotated separately using khipu[27] to identify isotopic patterns. All [M + H]+ features with correct isotopic patterns were combined into the 643 "certified" features. Missed feature numbers are 62 for XCMS and 22 for asari (further examined in Supplementary Fig. 6).

still returned very different numbers of features. It's a complex problem involving many components in software implementations. Just as in other –omics data, metabolomics features should be considered in a statistical context.

To give a visual guide to the problems, we illustrate a few typical EICs in Fig. 5a, In the "good" cases, almost any peak detection algorithm will work. In other cases, the consideration of noise level is critical. In high-noise EICs, more peaks can be obtained by lowering

stringency of parameters. An extreme example is shown in Supplementary Fig. 8b, where the number of peaks inflates quickly by varying parameters. We have to caution that sensitivity should not come at the cost of low data quality. At low noise levels, small peaks can be valid (Fig. 5a, upper right). A peak detection algorithm should ensure they are detected even at the presence of multiple big peaks.

The default peak detection algorithm in asari first estimates the baseline and noise level on a composite mass track (details in the Supplementary Information). The statistics drive the decisions on baseline subtraction, detrending or smoothing. Detrending is to regress out shifting chromatographic background, used sparingly in necessary tracks (e.g., Supplementary Fig. 8c). Noisy composite mass tracks are smoothed using simple moving average. Aggressive smoothing can lead to superfluous artifacts (the reason that raw data points have to be used for inspection) and is not recommended for general use. A composite mass track is then separated into segments of valid signals, and peak detection is performed on each segment. We use a local maximum search algorithm, with peak height and prominence requirements dynamically determined on the segment statistics. Prominence is the vertical distance of a peak top to its adjacent local minima, therefore important to control for fluctuating data points. The detected peaks are assessed by a set of quality metrics and retained only if the preset thresholds are met. In larger studies, asari's performance improves via the cumulative peak patterns in composite mass tracks.

To understand the vast differences in feature detection between tools (Supplementary Tables 1, 2), we need to ask how many good peaks are found by them. An intuitive measure of peak quality is "peak shape", here defined as the goodness of fitting to a Gaussian curve (other curves have been used but the impact on fitness scores is negligible). High quality peaks should have peak shapes close to 1. We recomputed the peak shapes from the result of each tool on the Yeast 2021 dataset (longer chromatography, small size of only three samples minimizing impact from mass alignment), and plotted them as a function of peak height in Fig. 5b. Because the reported retention time differs slightly between tools, this calculation extended 3 s on each side of the peak range, which led to slightly worse fitting for some small peaks. This extension accounts for the 504 features in red for asari, which, by default, does not return peaks of bad shapes (under 0.5). Other tools reported many more peaks with low quality peak shape: XCMS has 1243 bad peaks; MZmine (wavelets) 4035 (not shown); and MZmine (local minimum) has 9436 (all shown in red). These data indicate that the number of good features is relatively stable between tools. The large numbers of "extra" features reported by different tools are low-quality peaks, which are not reproducible and should not be used without additional evidence. Indeed, the overlap features between asari and others are stable at around 3000 (Supplementary Table 2). Figure 5b has close to 5000 good peaks (in black) for all tools, because not all features are uniquely matched, due to the correspondence problem discussed earlier. Summarizing results in Figs. 4, 5, larger numbers of reported features do not improve the real performance on benchmark peaks or good peaks.

## Asari includes a set of metrics to fully describe data quality
The discussion above shows the importance to safeguard the processing result through quality metrics. High numbers of bad peaks are not acceptable in real-world applications due to their negative impacts on downstream analyses. For biomedical studies, features of questionable quality should not be used for decision making. Because metabolomics is often used across multiple disciplines, it is important that data quality can be clearly assessed and that it is consistent with the quality of analytical chemistry.

SNR (signal-to-noise ratio) is often defined differently among tools. In asari, the noise for an elution peak is the average signal intensity of neighboring non-peak data points, 100 to each side; SNR is the ratio between peak height and noise (Fig. 6a). The mSelectivity metric

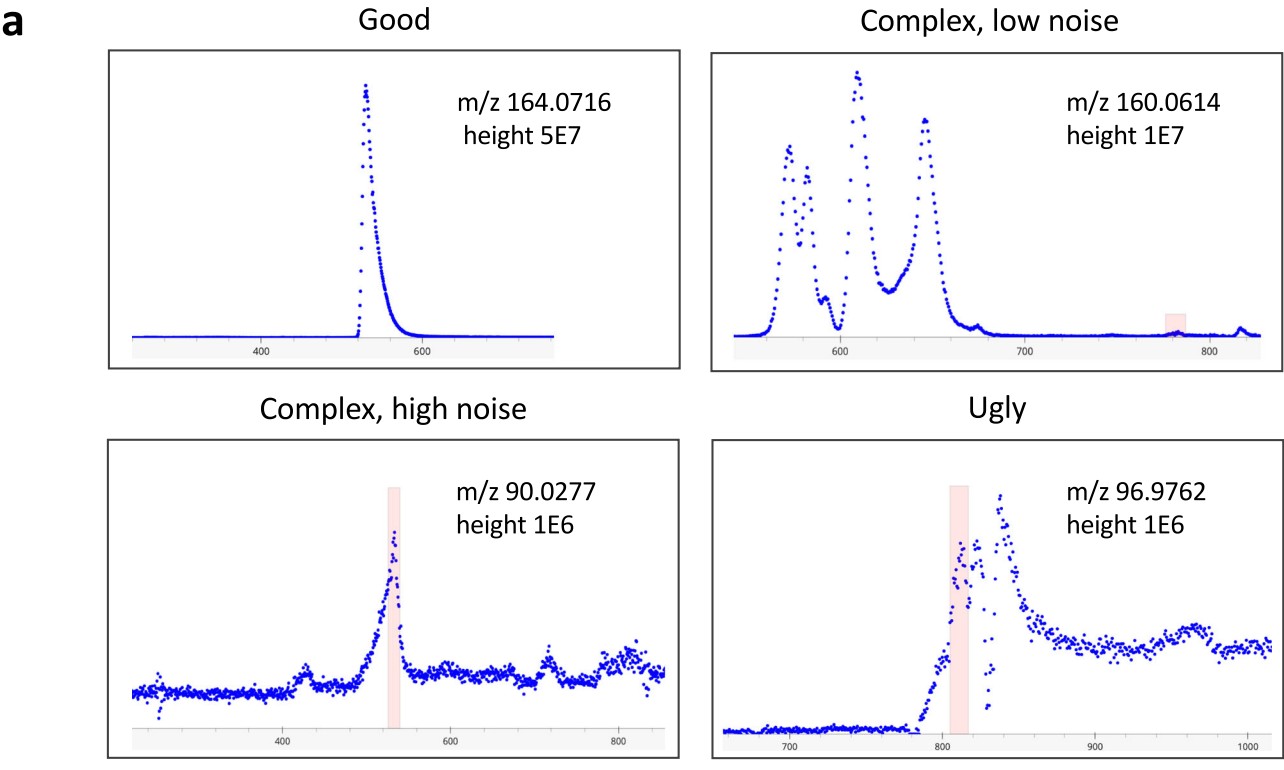

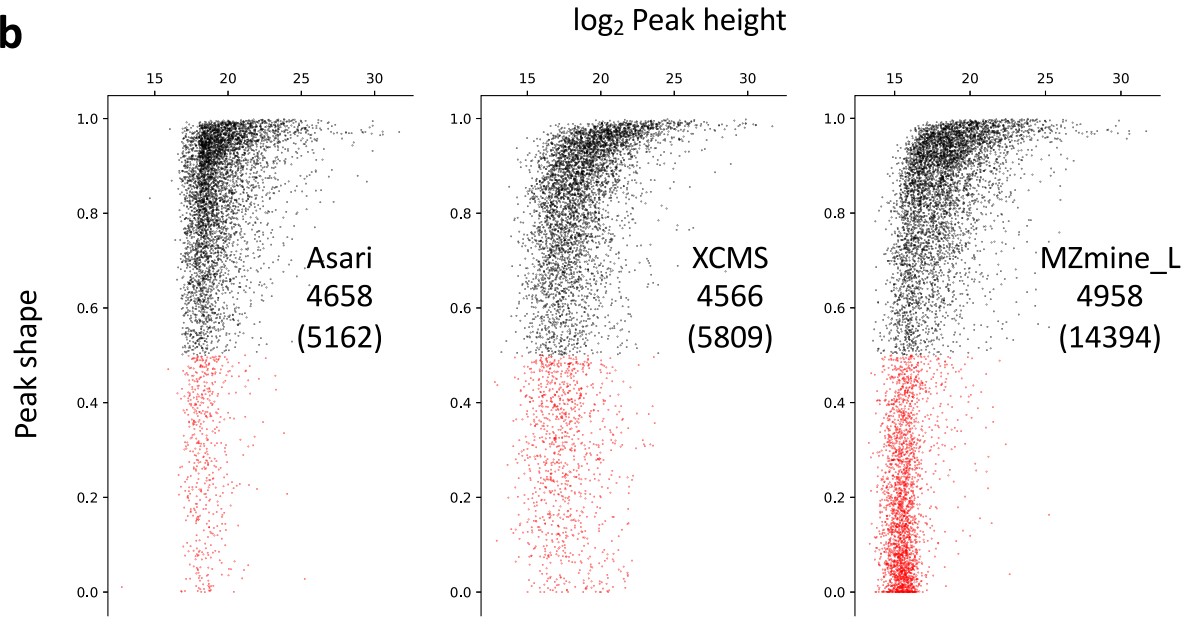

**Fig. 5 | The number of peaks has no significance beyond a core set of high-quality peaks. a** Examples of different types of EICs. Levels of maximum height, baseline and noise are used to inform algorithmic decisions involved in peak detection. Mass track view of all detected peaks for the top right example is provided in Supplementary Fig. 8a. **b** Re-computed peak shapes for features reported by different tools processing the Yeast2021 dataset (see Supplementary Table 2), plotted as a function of peak height. Each dot represents a LC-MS feature, black for Gaussian fitness score > 0.5, red for < 0.5. MS-DIAL is not included in this analysis because we did not find a straight forward method to export peak boundaries after alignment. For other tools, all features from each were mapped to a set of mass tracks generated by asari using low thresholds (minimal peak height 5000, peak shape 0.1, SNR 1.1). Almost all m/z values in the tools were matched to these mass tracks. For each feature, the retention time was padded 3 s on each side of peak boundary, to accommodate processing variations. The redundant features (within 5 ppm and 6 s) were first grouped in XCMS and MZmine results. Thus, the total feature numbers in this analysis are 5809 for XCMS, 10876 for MZmine (wavelets), and 14394 for MZmine (local minimum). The MZmine (wavelets) result has 6841 good peaks and 4035 bad peaks. They are not plotted here because too many low intensity peaks returned peak shape of 0 s in this approach. Source data are provided as a Source Data file.

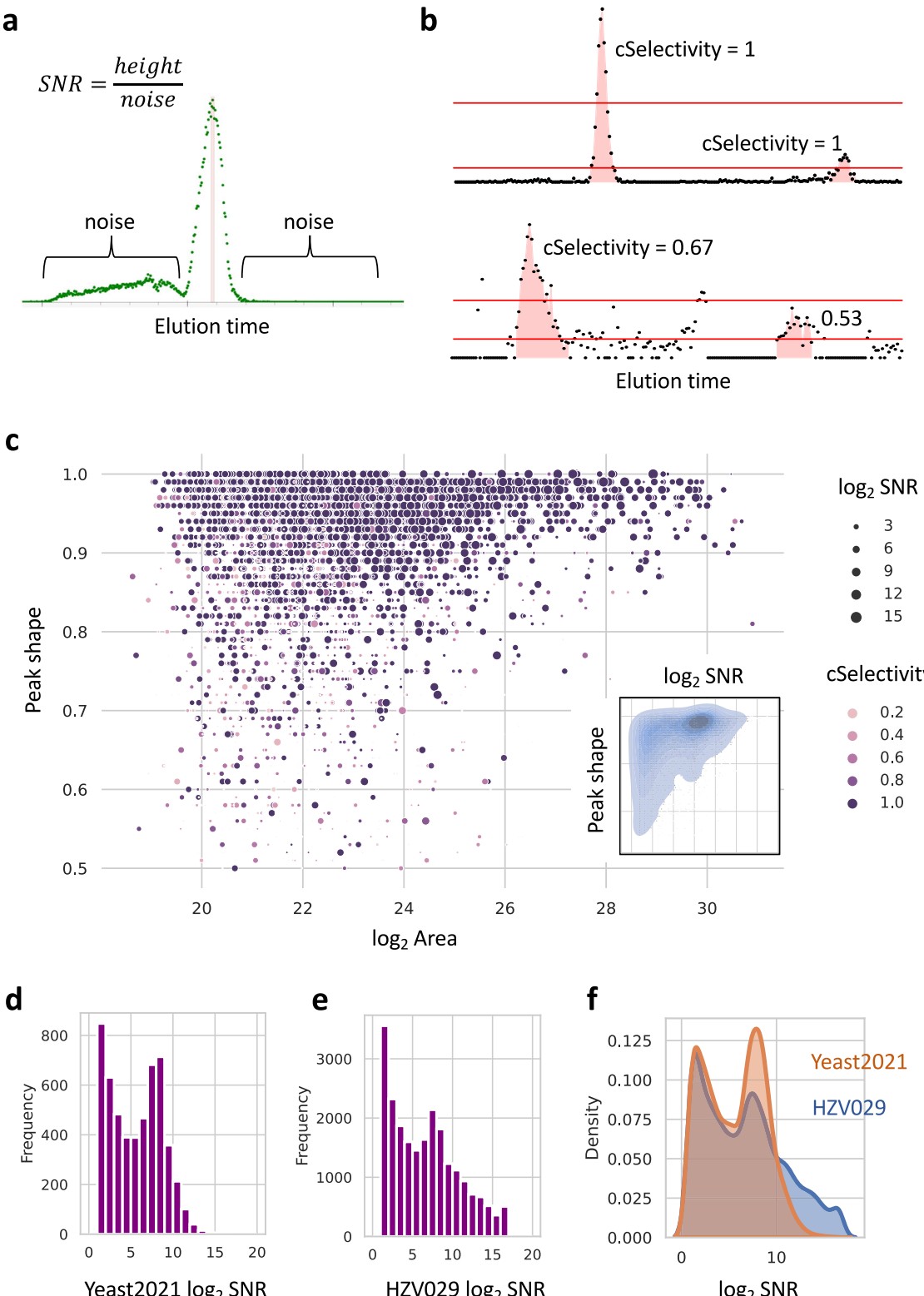

**Fig. 6 | Quality metrics in asari and applications to data review. a** Signal-to-noise ratio (SNR) in asari is defined by peak height divided by noise level. In a mass track (extracted ion chromatogram), all data points within the peak range are considered part of the peak. All other data points are considered as noise. The noise level is taken as average intensity of up to 100 nonpeak data points on each side of the peak. **b** Chromatographic peak selectivity (cSelectivity). After filtering the data by 1/2 of a given peak height, the cSelectivity of the given peak is defined by the fraction of the data points belong to any peak. cSelectivity is 1 when the chromatogram has no noise above the half height of any peak. **c** Overview of all features in the Yeast2021 dataset. Peak area (log2) and peak shape are x and y-axes; SNR and cSelectivity are coded by size and color. Contrast to data of lower quality is shown in Supplementary Fig. 9. The inset shows the kernel density of SNR and peak shape. **d, e** Histograms of SNR distribution in the Yeast2021 and HZV029 datasets, respectively. **f** Kernel density of SNR distribution in the Yeast2021 and HZV029 datasets. More features with high SNR indicate better quality in feature detection. Source data are provided as a Source Data file.

(Fig. 1c) is integral to mass track construction and mass alignment in asari. Similar for chromatography, cSelectivity is defined as how distinct chromatograhic elution peaks are (Fig. 6b). Together with peak shape, users can rely on them to routinely determine the peak quality. Global visualization of these three quality metrics in Yeast2021 data, in relation to peak size, is shown in Fig. 6c. This figure demonstrates the desired qualities of good peak shape, high SNR and high cSelectivity on asari features. The same plot can be used to weed out poor data quality, as illustrated in Supplementary Fig. 9a. The SNR distribution is an informative way to assess how much useful information is in a metabolomics dataset (Fig. 6d–f, Supplementary Fig. 9b). These quality metrics are exported with the feature tables in asari. In our own research lab, features of interest in untargeted data used to require inspection of peak quality. With asari and these quality metrics, the burden of manual inspection is mostly removed. Users can be confident of feature quality based on these metrics and own the decisions of filtering data.

### Feature quantification is evaluated favorably in asari

The quantification of each peak is usually based on the peak area, which is represented in asari by summing the intensity of each data point in a peak. The peak areas by XCMS and asari are generally agreeable (Fig. 7a, b). To further investigate the performance in quantification, we designed an experiment where human plasma and vegetable juice were mixed by varying ratios (BM21 dataset, Fig. 7c). Therefore, a majority of features are expected to have their peak areas correlated with the mixing ratios. Overall, 5581 features were matched between XCMS and

asari in the BM21 dataset. Their Pearson correlation coefficients to the mix ratios were computed, and the distributions are shown in Fig. 7d. Asari has more features with correlation coefficient >0.9, indicating that the quantification in asari is more useful than that of XCMS.

### Asari delivers significant improvement in computational performance

Computational efficiency is fundamentally important to−omics data. For MZmine and MS-DIAL, processing 100 samples often becomes challenging on a desktop computer. XCMS is considered more performant and is the default choice of cloud computing[13,25,26]. Here, the computational efficiency of asari is benchmarked against XCMS using multiple datasets of different sizes and platforms. Small studies can be processed by asari under a minute. Some studies of 100 - 200 samples take less than 10 min by asari using a single CPU core. Therefore, asari provides significant improvement of CPU time over XCMS by 1 - 2 orders of magnitude (Fig. 8a).

To test the scalability, we subset the SLAW data[13] using varying sample numbers. The CPU time and memory use was largely a linear function of sample numbers (Fig. 8b, c). The results indicate that the performance gap between XCMS and asari widens for larger studies. XCMS can also become more complicated if it goes beyond simple workflows or large studies are processed[13]. The full SLAW dataset of >2000 samples was processed in the previous study by XCMS on a cluster node of 15 CPU cores in 7-12 h. This same dataset was processed in ~1 h on a regular laptop computer using asari. This enables

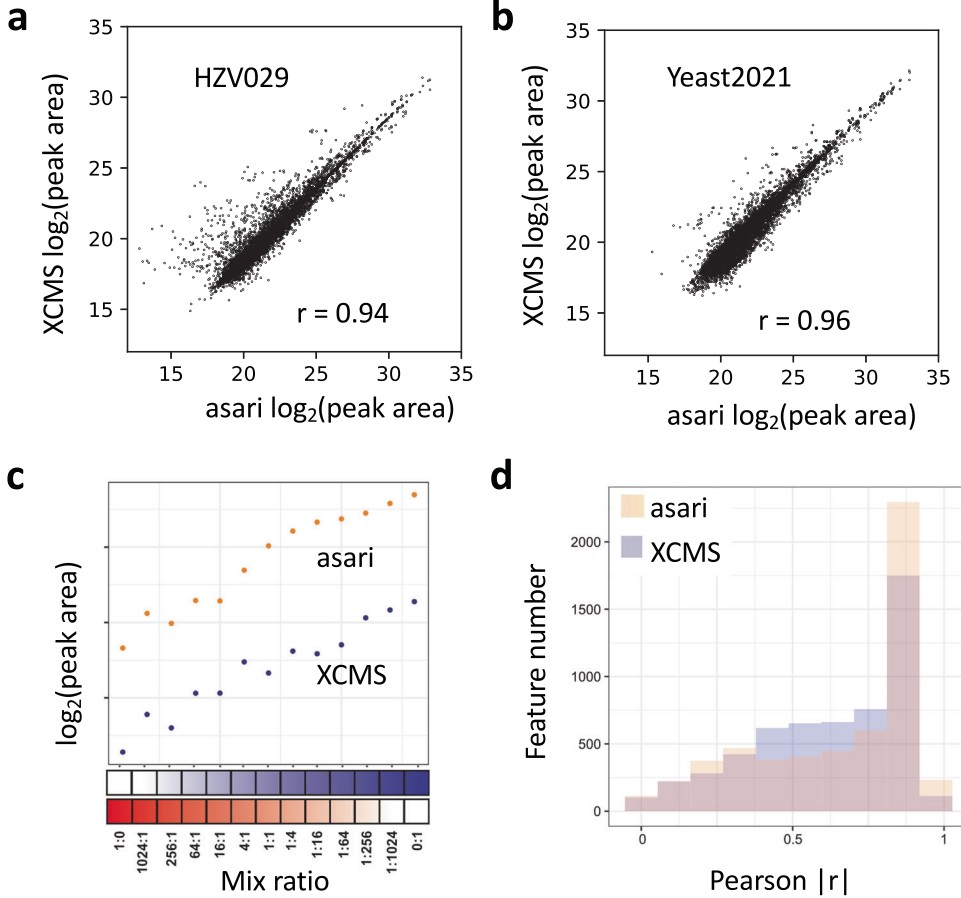

**Fig. 7 | Evaluation of feature quantification.** Scatter plot of the $log_2$ peak areas of common features between asari and XCMS, on three samples in HZV029 dataset (**a**) and Yeast2021 dataset (**b**). The r values are based on Pearson correlation. **c** Design of the BM21 dataset, by varying mix ratios between human plasma and vegetable juice. A well quantified metabolite is expected to show good correlation between the mixing ratios and the reported peak areas, as exemplified by the feature on top (m/z 189.1232, 159 s). Asari calculates peak area differently from XCMS, resulting in higher values in Orbitrap data. **d** Overall quantification results in the BM21 dataset, shown as feature numbers binned by Pearson correlation coefficients between peak areas and sample mixing ratios. Source data are provided as a Source Data file.

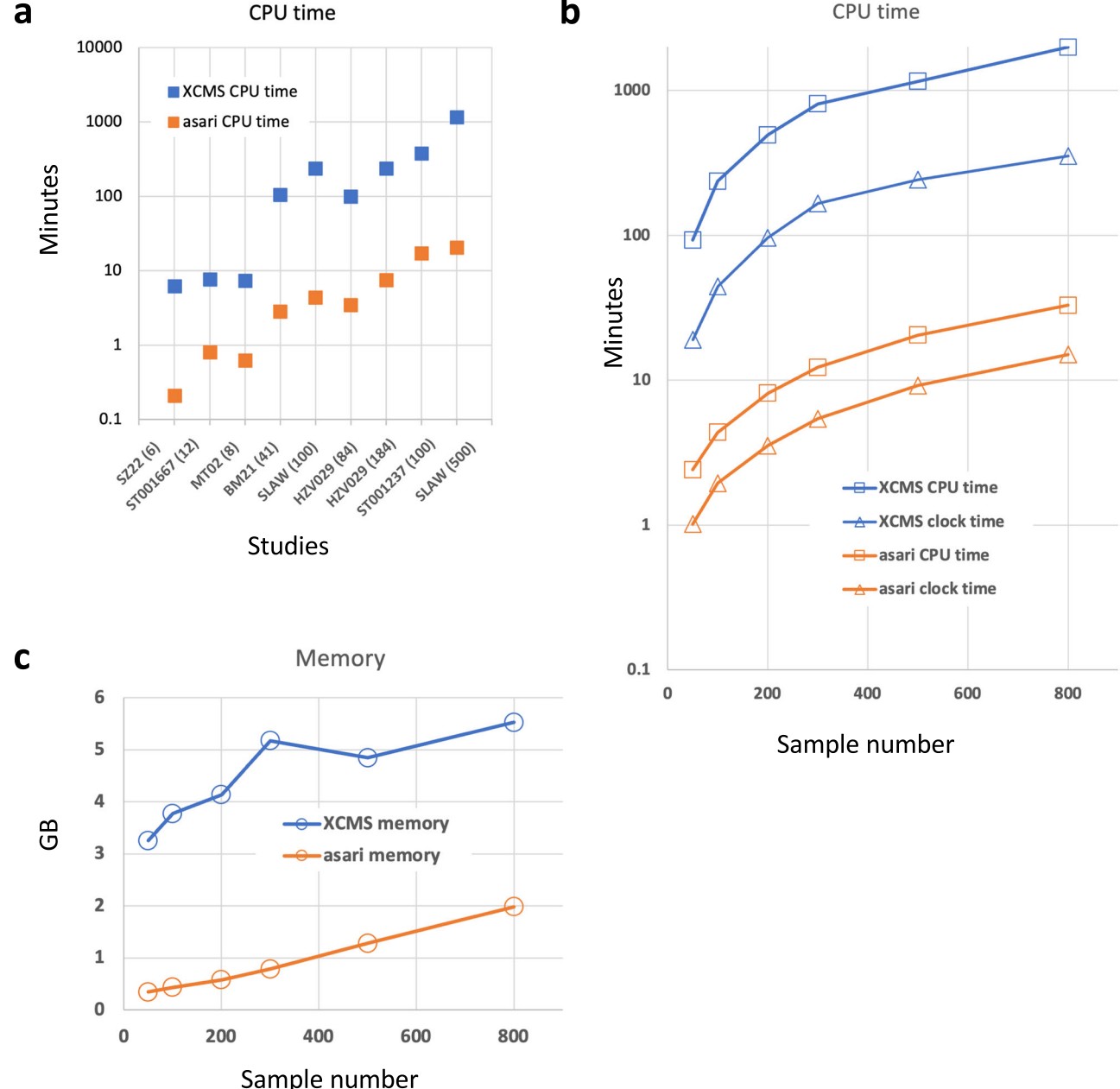

**Fig. 8 | Evaluation of computational performance. a** Computational performance in user CPU time (equivalent to single core) by asari and XCMS on different datasets (sample numbers show in parentheses on *X*-axis). CPU time and wall clock time (**b**) and memory (**c**) used by asari and XCMS on the SLAW dataset using varying number of samples. Y-axis is in $\log_{10}$ scale for CPU time (**a**, **b**). Source data are provided as a Source Data file.

expeditious metabolomics data processing on cheap hardware and makes very large studies feasible.

**Reusable data structures and code, a bridge to data science**
The asari software is built on a set of transparent data structures (Fig. 2c). Mass tracks are an extension to the concept of extracted ion chromatograms and serve to simplify m/z alignment and navigation. A MassGrid records the alignment of mass tracks across samples. A feature is defined at the experiment level, and elution peaks are defined at sample level. Mass tracks across samples are superimposed and summed, to become composite mass tracks. The composite mass tracks are representative of all samples. A metabolite may have multiple degenerate features due to isotopes, adducts, neutral loss and fragments, which are grouped into an "empirical compound" via another package khipu[27]. An empirical compound is a computational

unit for a tentative metabolite, since the experimental measurement may not separate compounds of identical mass (isomers). Asari explicitly links mass tracks, peaks, features and empirical compounds, so that each processing step can be traced and verified. Examples of data structures are provided in Supplementary Methods and in our code repositories. These data structures are JSON compatible and exposed, so that advanced users and developers can reuse them easily.

We designed asari with the goals of code reusability, easy deployment, easy maintainability, cloud friendliness, and scalable performance. The software is open source on GitHub and available via standard Python package management tools, which can be integrated seamlessly with cloud deployment. Docker images and build recipes are available. Subcommands are designed to perform common tasks, such as batch processing, analyzing experimental parameters, targeted extraction, and visualization via the dashboard. The software

modules can be ported for other tools or scripting, as demonstrated by Jupyter notebooks in the code repositories.

Better integration of metabolomics with biomedicine requires metabolomics data processing tools that are accessible to general data scientists, not limited to chemists. Previous tools are burdened with complicated parameters while asari requires almost no tunable parameters. To meet the diverse demands of advancing science, the software has to be interoperable, and interface with the rich tools in general data science. The above features of asari are designed to bridge the gap to data science and fit easily into automated pipelines.

## Discussion

Reproducibility in data processing has been a roadblock for metabolomics. In this study, we attribute the inconsistency in previous software tools to poor mass alignment and a lack of quality metrics. The widely used wavelet algorithm shows no advantage in this study, but costs unnecessary complexity. We have developed the asari software, which compares favorably to other software in feature detection. Arguably, the detection performance of the previous tools may be improved by further parameter tuning. But this study highlights that the number of high-quality features is stable, and increasing feature numbers also risks increasing the number of undesired false positives.

Using a statistically-minded approach, asari does not require users to supply any tunable parameter than mass resolution, where the default value of 5 ppm rarely requires changing. Advanced users can opt to modify parameters and workflows. Previous software tools often deteriorate in feature correspondence in larger studies. On the contrary, the processing quality in asari increases with larger studies, because it utilizes recurrent patterns in feature detection. Asari has a short history and has been mostly tested on the Orbitrap platforms. The settings will need further optimization for other platforms. The software does not work with low resolution data. The efficiency is not optimal on data from longer chromatography. The default parameters are designed to work with a broad range of data, but data of very high noise may need customized processing. Future research and development is clearly required to improve the tool. For example, additional methods of retention time alignment can take advantage of spike-in standards. Lipidomics and xenobiotics can benefit from additional specific modules. Metabolomics platforms and methods are diverse and always evolving. No single group can address all the computational needs. We will maintain an open development model and welcome community involvement. Asari can be easily chained with or incorporated into other tools.

The mass tracks in asari are an operational unit to ensure correct information retrieval, not necessarily resolving close m/z values. This design does not interfere with annotation when the existence of multiple compounds is known—they are found on a mass track within a preset error range. There have been debates on the use of absolute amu or relative ppm for mass resolution in data processing[28]. Asari uses ppm currently. But the ppm is still a practical approximation. In the same data, relative mass errors in ppm can get larger in the higher m/z range[29]. Thus, the absolute amu can be considered a 0 order and the ppm 1st order of a polynomial model of mass accuracy, in relationship to m/z. But a 2nd order polynomial model may be required to cover the full m/z range more accurately. Currently, asari keeps and retrieves data within a preset resolution in a predictable way. For more specialized applications, researchers may take into consideration these limitations. The borderline cases are often a tradeoff between sensitivity and true discovery rate. The design of asari favors the latter because it has less impact on data analysis and sensitivity should have the support of annotation.

In summary, the development of asari has significantly contributed to the reproducible data in metabolomics, using a full set of linked and transparent data structures in all processing steps. This allows developers to trace, debug and optimize the process into the future. The end users can navigate and verify features by interactive visualization of extracted ion chromatograms in asari dashboard. Asari's improvements in computational performance and tractability will foster the increased use of metabolomics in biomedicine.

### Reporting summary

Further information on research design is available in the Nature Portfolio Reporting Summary linked to this article.

### Data availability

The BM21 and HZV029 datasets have been deposited and are available at Metabolomics Workbench (https://www.metabolomicsworkbench.org) under the study IDs ST002454 [https://doi.org/10.21228/M86Q7N] and ST002233 [https://doi.org/10.21228/M8SD86], respectively. The datasets MT02 and SZ22 and the list of verified NIST SRM 1950 features are available at https://github.com/shuzhao-li/data. The large SLAW dataset[13] is available from MassIVE (https://massive.ucsd.edu) under study ID MSV000086486 [https://doi.org/10.25345/C5GJ4W]. The Yeast201 data[21] is available from MassIVE under study ID MSV000087434 [https://doi.org/doi:10.25345/C5WV53]. The other public datasets used in this work are available under study IDs ST001667 [https://doi.org/10.21228/M8NM4H] and ST001237 [https://doi.org/10.21228/M8NQ4J] on Metabolomics Workbench. Source data are provided with this paper.

### Code availability

The asari source code is available at GitHub, https://github.com/shuzhao-li/asari, and as a Python package via https://pypi.org/project/asari-metabolomics/. Jupyter notebooks used for data analysis in this paper are provided at GitHub, https://github.com/shuzhao-li/data.

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

## Acknowledgements

This work was in parted funded by NIH grants (to SL) U01 CA235493 (NCI) and R01 AI149746 (NIAID). We thank Drs. Jeff Xia and Joshua Mitchell for discussions of the writing and Dr. Sara Cassidy for the editing of this manuscript.

## Author contributions

S.L. designed the study, wrote the asari software and the manuscript. A.S. and S.L. performed data analysis and software testing. M.T. performed the experiments of HZV029, MT02 and BM21. S.Z. performed the experiment of SZ22. Y.C. contributed to asari visualization and documentation.

## Competing interests

The authors declare no competing interests.
