## [Peer Review File · Nature Communications]

Reviewers' comments:

Reviewer #1 (Remarks to the Author):

The authors are presenting a novel data processing workflow for LC-HRMS data "asari". The approach proposes two key advances. First, the peak detection is performed on a merged overall sample instead of sample-wise; this has two desirable effects: 1) it cuts down on computation time, since peak detection needs to be done on a single "sample" only, and 2) perhaps even more significant, it completely sidesteps the error-prone and difficult step of aligning peaks between samples. Second, processing is trackable, tracable and reproducible.

Concerning the "composite" peak detection, the approach is quite clever. I could see it find broader application if it manages to overcome the pervasive inertia of the field regarding software.

I would like the authors to address a few questions here:

* They claim that a peak can be detected even if present in single samples. However, the process itself seems to disfavor this, as peak detection is filtered by a mean intensity value, so low-intensity features should be diluted out in the "mixture". Can you show data that convinces me that this works in practice? The algorithm is compared to XCMS on intense features and on frequent features, which are exactly the ones easy to detect with this algorithm.

* What about retention time shifts over longer runs? The authors describe a retention time correction, but they don't show how this actually works, how well it works, and where it fails.

* What about isomers close in retention time? How good is it at keeping them apart, or more generally, what is the outcome of the algorithm for those?

Overall, the evaluation is a bit softball as it evaluates relatively easy cases. For a publication on this level I want to see how it handles the tough stuff.

A lipid dataset might be an interesting test case here, since there are lots of isomers that need to be correctly matched across samples. There are now also datasets of manually annotated peaks that could be interesting to evaluate, e.g. <https://doi.org/10.3390/metabo10040162>

These comments are perhaps a symptom of a wider issue, namely that after reading the paper I have no idea how the algorithms actually work. The authors perform peak detection by detecting peaks, perform RT alignment by aligning retention times, etc without ever describing the algorithms. The complex process of annotation grouping is literally only described by "Annotation".

Similarly, they provide novel transparent, tracable, trackable linked data structures without actually showing what these data structures are. Processing with XCMS is also "reproducible", tracks processing steps, and allows to trace features back to raw files, so just claiming that there are somehow data structures that make everything better does not convince me.

The authors also describe quality metrics mSelectivity, cSelectivity, dSelectivity. cSelectivity is the only one actually described, as the fraction of data vs noise points above half peak height. (Note the typo on figure caption 1D.) But what is a "data" or "noise" point? The other metrics are not described at all, neither are they justified as why they are good metrics for anything or what "good" or "bad" values are.

I think this sums it up well - the method has promise, but unfortunately there is not enough detail to truly understand what happens.

Reviewer #2 (Remarks to the Author):

Li and colleagues introduce a new preprocessing tool for LC-MS data called Asari, and compare its performance with XCMS. The results show a significant speed and memory usage improvement and, it seems to show a better peak extraction performance, thus improving feature quantitation performance. While tools that improve processing speed and quantitation are needed and are welcome, I do not believe that these improvements have the necessary impact to be published in Nature Communications. I also have the following concerns and suggestions:

- The authors compared the performance of Asari with XCMS via different approaches, but those include a low number of metabolites (features compared). I suggest that the authors analyze a large set of standards to identify more metabolites in real samples. Commercial libraries or kits like IROA or Biocrates now allow the identification of ca. 600 metabolites. Then, check if the in-source fragments, isotopes and adducts observed in the well-concentrated standards analyses are also found in real samples (generate a large, ground truth feature list). This will allow you to compare how well the two software work. This would also enable performing correlations among adducts, in-source fragments and isotopes, to assess the quantitative performance. Log transformation and other scaling transformations should be used to avoid the high abundance of particular features introducing a positivity outlier bias that increases the correlation values.

- The authors just compare Asari with XCMS. I believe that a comparison with other established software like mzMine or MS-DIAL could demonstrate the advantages of Asari.

The authors state: "asari has significantly contributed to"... "linked and transparent data structures in all processing steps". XCMS also shares transparent data structures in all processing steps. Therefore, the scalability claim of Asari can also be attributed to XCMS, and even mzMine, which are both in Github.

- The methods are not very well explained. The authors give a global picture of the algorithm but there is a lack of details about how the algorithm operates.

- Fig 2G, why the CV's are not calculated for XCMS?

Point-by-point response to reviewers' comments

We agree with the reviewers that this work can benefit from more technical details. As such, the manuscript is completely rewritten.

The main text is expanded from 1500 words to 5000 words, from 3 figures to 8 figures. Over 80% of the text is new. Figures 1A-B, 1D, 2A, 4, 5, 6A, and 6C-F are new. Figures 7 and 8 are refreshed with slight version changes. Figure 3 is extended from a previous supplemental figure.

In the revised manuscript, we describe detailed investigation of issues in the current tools (**Figures 1, 5, S4; Tables S1, S2**), including prevalent problems in mass alignment and lack of safeguard against questionable features at noise level. For -omics technology, sensitivity of feature detection is not meaningful if it leads to too many false positives. We would like to point out that the real issue is not more peaks but reliable peaks (**Figure 5**). The evaluation of feature detection is now focused on three datasets of large numbers of verified features.

Reviewer #1

1. They claim that a peak can be detected even if present in single samples. However, the process itself seems to disfavor this, as peak detection is filtered by a mean intensity value, so low-intensity features should be diluted out in the "mixture". Can you show data that convinces me that this works in practice? The algorithm is compared to XCMS on intense features and on frequent features, which are exactly the ones easy to detect with this algorithm.

Peak detection in asari is not filtered by a mean intensity value. The sum of all samples is used for feature detection (**Figures 2B, S2**). Quantification is decoupled from the detection step, by looking up intensity values in individual samples after a peak is defined in the composite map. Peak detection is now extensively described in this new version.

We have detailed discussions in this new version that extra information from mass alignment improves the confidence in calling peaks in a single sample. Anyone using asari can see these features of low frequency, including in a single sample, in the feature tables. It is easy to use the `viz` subcommand to launch the Dashboard, and visually inspect any feature of any frequency.

We have updated the evaluation datasets (all new **Figure 4**). The original figure on high-intensity and frequent features is left out, because the new Figure 4 and supporting figures are deemed adequate.

2. What about retention time shifts over longer runs? The authors describe a retention time correction, but they don't show how this actually works, how well it works, and where it fails.

We demonstrate that retention time correction is not the most critical problems in the current tools. Mass correspondence and peak detection are the critical problems. Our results on complex chromatograms (e.g. **Figures 3B-D, 5A, S3, S4**) show that the software works well. The retention time correction method in asari is based on LOWESS regression, and is described in this new version. The decision of LOWESS regression was based on our experience, but it was the same choice made in the SLAW project (Delabriere et al, 2021).

We designed the software implementation to be “fail-safe”: a) the LOWESS regression is controlled on both boundaries, not to have spurious errors; b) because the mass alignment has high fidelity, failed retention time alignment will still have peaks on the same mass track; and c) some QC samples (blanks, external control samples) don’t align and will still be processed and corresponded without retention time correction.

3. What about isomers close in retention time? How good is it at keeping them apart, or more generally, what is the outcome of the algorithm for those?

With the three benchmark datasets in this new version, we manually examined all results and asari did not miss a single good peak. In our algorithm, the dynamic peak prominence is able to resolve close peaks (e.g. **Figure 3B-D**; and the peaks missed by MS-DIAL but detected by asari, **Figure S4**).

4. A lipid dataset might be an interesting test case here, since there are lots of isomers that need to be correctly matched across samples. There are now also datasets of manually annotated peaks that could be interesting to evaluate, e.g. <https://doi.org/10.3390/metabo10040162>

We thank the reviewers for the idea of extending benchmark data. We do not claim that asari is superior in lipidomics data. The focus in this paper is fixing issues in provenance and performance for metabolomics. The recommended dataset, (Muller et al, 2020. "A data set of 255,000 randomly selected and manually classified extracted ion chromatograms for evaluation of peak detection methods." *Metabolites* 10.4 (2020): 162.), however, has multiple issues.

Muller et al (2020) proposed 5,000 peaks for benchmarking. The chromatogram construction is biased because it was based on MZmine. The figures for visual inspection were plotted via XCMS using ROIs. We point out in our paper that ROIs are not equivalent to chromatograms and extra data points cause confusion. Another critical issue is that the study selected overwhelmingly low-intensity peaks, many at noise level – we only found 345 peaks above 1E6. A reflection of these issues is its own reproducibility. Once we reprocessed the dataset following the same method in the study (identical parameters in MZmine) but excluding QC samples, only 3,450 of the published 5,000 peaks were detected. Therefore, we do not consider this a suitable dataset to test the full processing workflow.

We instead added the dataset by Chen et al ("Metabolite discovery through global annotation of untargeted metabolomics data." *Nature methods* 18.11 (2021): 1377-1385), besides extending our own datasets.

5. These comments are perhaps a symptom of a wider issue, namely that after reading the paper I have no idea how the algorithms actually work. The authors perform peak detection by detecting peaks, perform RT alignment by aligning retention times, etc without ever describing the algorithms. The complex process of annotation grouping is literally only described by "Annotation".

We apologize that the initial version was written in a short format. The technical documentation has been provided on GitHub. But as the reviewer correctly pointed out, this article should describe sufficient technical details, especially given that there's a paucity of them on this topic in general literature.

The article is now rewritten to include these details, extended from 1500 to 5000 words, 3 to 8 figures, as mentioned above.

However, annotation is complex, as stated, and should be a separate research topic from preprocessing. We have submitted the annotation manuscript elsewhere, which is referred in this manuscript (under minor revision at *Analytical Chemistry*, cited as BioRxiv URL).

6. Similarly, they provide novel transparent, traceable, trackable linked data structures without actually showing what these data structures are. Processing with XCMS is also "reproducible", tracks processing steps, and allows to trace features back to raw files, so just claiming that there are somehow data structures that make everything better does not convince me.

We have added to this version of manuscript extensive text and illustrations of the concepts. Examples are added to **Methods**. Detailed examples are also provided in the GitHub repositories, via Jupyter notebooks, READMEs, scripts and docstrings.

It is important to clarify on `XCMS is also "reproducible"`. Backtracking in software relies on explicit links of objects. Plotting ROIs in XCMS does not obey provenance, because ROIs are not the same as EICs and a computational step is skipped in between. Expert users can verify data quality in XCMS, but automated software testing is very hard, which has been a problem for progress.

7. The authors also describe quality metrics mSelectivity, cSelectivity, dSelectivity. cSelectivity is the only one actually described, as the fraction of data vs noise points above half peak height. (Note the typo on figure caption 1D.) But what is a "data" or "noise" point? The other metrics are not described at all, neither are they justified as why they are good metrics for anything or what "good" or "bad" values are.

We have added all details and a dedicated **Figure 6**. “mSelectivity” is added as a section in **Methods**. The “dSelectivity” is now left out because it’s for a different application.

Reviewer #2

1. The authors compared the performance of Asari with XCMS via different approaches, but those include a low number of metabolites (features compared). I suggest that the authors analyze a large set of standards to identify more metabolites in real samples. Commercial libraries or kits like IROA or Biocrates now allow the identification of ca. 600 metabolites. Then, check if the in-source fragments, isotopes and adducts observed in the well-concentrated standards analyses are also found in real samples (generate a large, ground truth feature list). This will allow you to compare how well the two software work. This would also enable performing correlations among adducts, in-source fragments and isotopes, to assess the quantitative performance. Log transformation and other scaling transformations should be used to avoid the high abundance of particular features introducing a positivity outlier bias that increases the correlation values.

We thank the reviewers for this valuable suggestion. We have now extended the feature numbers in benchmarking to 402, 314 and 653 in three datasets (**Figure 4**).

This work focuses on preprocessing, i.e. confidence in extracting features, but identification is a separate problem. The commercial libraries like IROA are not tissue or platform specific. In our experience, they did not produce more than 300 identified metabolites in an experiment. We have considered the Muller et al dataset recommended by Reviewer #1, which however was not suitable, as discussed above.

Another dataset we looked into was from Li et al. "Comprehensive evaluation of untargeted metabolomics data processing software in feature detection, quantification and discriminating marker selection." *Analytica chimica acta* 1029 (2018): 50-57. The data in Li et al (2018) are based on mixtures of chemical standards, and have a very different signal distribution from common metabolomics data. The critical issue we encountered is errors in mzML conversion and parsing, therefore it cannot go through the same workflow as others in our paper.

We were able to include the dataset from Chen et al ("Metabolite discovery through global annotation of untargeted metabolomics data." *Nature methods* 18.11 (2021): 1377-1385), which contains 314 verified metabolites. Together with two extended datasets from our lab, we believe they give a fair assessment of software performance.

The quantitative performance is now in Figure 7, based on log scales. The correlations among adducts, in-source fragments and isotopes are an interesting idea, but there are also ongoing debates (e.g., Mahieu, et al. 2016. Defining and detecting complex peak relationships in mass spectral data: the Mz.unity algorithm. *Analytical chemistry*, 88(18), pp.9037-9046). Since quantification is not among the top issues discussed in this work, we prefer to keep it simple.

2. The authors just compare Asari with XCMS. I believe that a comparison with other established software like mzMine or MS-DIAL could demonstrate the advantages of Asari.

We have now included MZmine (v2 and v3) and MS-DIAL in comparisons of feature detection and quality (**Figures 1, 4, 5, S4; Tables S1, S2**).

They are not included in comparisons of computational performance, because

- a) XCMS is much faster than MZmine and MS-DIAL;
- b) larger number of samples becomes less feasible, even for XCMS;
- c) MS-DIAL is essentially for MS Windows only.

OpenMS was not included in the comparisons. It is still primarily a proteomics tool and there is no consensus workflow for metabolomics.

3. The authors state: "asari has significantly contributed to"... "linked and transparent data structures in all processing steps". XCMS also shares transparent data structures in all processing steps. Therefore, the scalability claim of Asari can also be attributed to XCMS, and even mzMine, which are both in Github.

The availability of source code does not guarantee the code being reusable and reproducible. It is more important to have a robust design that can test each step as in automated software testing. The software backtracking mechanism does not appear to be complete in XCMS, as we explained in the answer to Reviewer 1's comment #6. The data structures in asari are centered on JSON, which is the default standard for web applications and language neutral. They are more portable than R objects.

It's very challenging to process large studies by the previous tools. This computational bottleneck is now removed by asari.

4. The methods are not very well explained. The authors give a global picture of the algorithm but there is a lack of details about how the algorithm operates.

The article is now rewritten to include these details, expanded from 1500 words to 5000 words, from 3 figures to 8 figures.

5. Fig 2G, why the CV's are not calculated for XCMS?

The XCMS result has many missing values, leading to poor CVs. The old Fig 2G is now left out because the focus has shifted in the new version.

REVIEWER COMMENTS

Reviewer #1 (Remarks to the Author):

****General:****

- 1) I appreciate the much more extensive writeup in the new version.
- 2) I believe I kind of understand what the algorithm is doing now. However I still think the authors are not describing the algorithm at the necessary level of detail.
- 3) I have some issues with the writing. The authors should refer to e.g. XCMS or ADAP manuscripts for the precision and clarity expected.
- 4) With regards to the concept of "mSelectivity", I believe the authors are conceptually confusing a few things and need to clear things up a bit.

In summary:

- I think the algorithm is great and deserves publication. (I also think the authors should acknowledge where potential weaknesses are and be realistic that it doesn't solve all problems in the world.)
- However, the authors should bring this to the quality of a typical computational manuscript, get rid of buzzwords and hyperbole, and clear up the writing. It seems that the article is written from a quite "internal" perspective from a group that has worked on this for a while. It would benefit of the perspective of someone external, ideally someone who has experience with computational/algorithm manuscripts.

As I understand it, the algorithm works as follows:

****MassGrid detection****

- 1) Mass track detection: For each sample, ad-hoc m/z bins (or clusters) found by the equivalent of single-linkage clustering with a fixed height on the m/z axis (L137-138)

- 2) Alignment of mass tracks across samples for m/z correction by an unclear method (L140-146)
- 3) "Mass tracks" to "Aligned mass tracks": Merging of (corrected) mass tracks across samples. It is not clear if this again uses the equivalent of single-linkage clustering.
- 4) Per-sample RT correction by lowess on selected features from aligned mass tracks (L164-L167)
- 5) "Aligned mass tracks" to "Composite mass tracks": RT-corrected aligned mass tracks are summed to a "composite mass track". It is unclear how one arrives at discrete time bins over which to sum intensities, or how this is otherwise done. These "composite mass tracks" together are then also called a "MassGrid". (L162-171)

****Peak detection per composite mass track****

Elution peak detection seems to play a very important role, but is very poorly described.

- 6) There is some baseline and noise estimation (L254) It is unclear whether this uses the SNR definition from L290-292 or what it does. "The statistics then drive the decisions on baseline subtraction, detrending or smoothing", with no description of the decisions.
- 7) A mass track is then separated into segments of valid signals (L259-260), how?
- 8) Peak detection per segments with local maximum search, the rest remains vague

Finally, computation of peak shape, SNR and "cSelectivity", which serve for later inspection but don't have an influence on the processing itself.

I need to painfully piece together this information, and it remains quite incomplete. To me, it's quite clear: either an existing algorithm is used, then it should be referenced; otherwise, a new algorithm is used, and it should be described, e.g. in pseudocode. ****From the description, I should be able to write a bad implementation that does more or less the same thing****, some corner cases perhaps excluded but possibly mentioned. For an example of a decent and precise algorithm description, see e.g. the ADAP paper <https://doi.org/10.1021/acs.analchem.7b00947>, XCMS paper <https://doi.org/10.1186/1471-2105-9-504>, or the SI of <https://doi.org/10.3390/metabo6040039>.

Finally, ****concerning the writing****: There is a lot of hyperbole, big statements and buzzwords in the paper. Further, some words are misused, like "cannot" instead of "should not" or a more idiomatic construct. The authors should revise this, less is more here.

Just some examples. This is by far not exhaustive.

- "Asari provides verifiable elution peak detection". What does verifiable even mean here?
- "A set of metrics fully describe data quality"
- "Asari delivers a new generation of computational performance" (aside from the hyperbole, what does this even mean?)
- "Asari marks a clear transition"
- "a bridge to data science", "bridge the gap to data science"
- "Computational processing cannot overstate the quality of analytical chemistry". What is this supposed to mean?
- "Since asari considers all [...], confidence can be established at high level of sensitivity": What "confidence" are they talking about?
- "By linking features and raw data through composite map, asari produces an efficient solution of automated testing, debugging and verification". ??? What is tested automatically? What is debugged? What is verified? (Clearly the authors don't mean automated testing of the code, as the code repository doesn't seem to contain any unit tests.)

****Regarding the mass track concept in general****

The key ideas of the paper are "mass tracks" and "mSelectivity". The notion is that

- 1) masses closer to each other than a preset resolution, described by mSelectivity, cannot reliably be distinguished
- 2) combining masses into mass tracks provides a reproducible basis for downstream analysis
- 3) peak detection on combined mass tracks instead of single samples is both more reproducible and more efficient.

While 2 and 3 are correct and great ideas, 1 confuses two things. Clearly, 528.2374 and 528.2399 (4.7 ppm apart) can easily and reproducibly be **distinguished** on the mass spectrometer (in two separate injections, or with different retention times), even if they might not be **resolved** in a single scan even on an Orbitrap. Peak picking with e.g. XCMS will reliably determine the two masses

separately, which is important for formula assignment. Even TOFs with relatively low resolutions (20-30k) can determine the mass apex quite accurately.

I think it is a great idea to merge masses to mass tracks for processing. However, this should not be taken to mean that such masses are in fact all the same. The "mass track" method is, in a way, artificially recreating the analytical problem of "coalescence", which occurs for close masses in a single scan in Orbitraps (Gorshkov 2012 Rapid Commun Mass Spectrom., Kaufmann 2018 Rapid Commun Mass Spectrom.)

I would suggest, after peak detection, to go back and calculate the "real" m/z for a peak for the detected feature.

****Individual comments****

L120-123 There is no "manual" merging in XCMS as far as I am aware so that discussion is out of place

L140: "locks the origin of peaks in computational processes"

L141: Is the combination of sample mass tracks to composite mass tracks again equivalent to single linkage clustering, or is something else done there? I.e., if track A1 and A2 in sample A are 6 ppm apart, and track B3 in sample B is in between (after alignment), do they all get merged to a single composite mass track?

L145-146: "We can already use $^{13}C/^{12}C...$ " Is this actually done, and if yes, how? Or is this a future possibility? The methods don't show this.

L172: "Previous tools usually require the presence in multiple samples to call a feature, because of their high error rates": This is simply not true. Users are free to set whatever threshold they wish for feature sample presence.

L181ff: I tried the viewer. It seems that one can only look at composite mass tracks, not at individual samples? This would be helpful to understand if e.g multiple peaks could come from an RT alignment problem.

L187: "the computational complexity increases quickly [...] if all intermediate steps are recorded" This doesn't make sense, the computational complexity stays exactly the same. Maybe the memory complexity changes if stuff is kept in RAM, which was probably the case in MZMine 2 (and is quite unnecessary, I agree). WRT computational complexity, are the authors aware of things like big-O notation?

L204ff: The SI show that extremely different parameter sets were used for the different workflows, for example $1e5$ (!) minimal peak height for asari, $1e3/1e4$ threshold for ADAP/MZMine, $1e3$ noise filter for XCMS, $5e3$ for MS-DIAL. Is there any rationale for this? This makes the results from Fig4 look somewhat arbitrary. Given that there is an obvious sensitivity parameter, could one construct something like a ROC curve?

L238: "as so many confusions are related to peak detection as well as the coverage of metabolomics". What is this supposed to mean?

L239-241: While ADAP uses wavelets too, the overall algorithm is obviously still quite different. The statement makes no sense. If two algorithms are the same, they will give the same result with the same parameters.

L247: I think we can agree that Fig 5 top-right is a complex case; what does Asari detect there?

L255: The statistics drive the decisions... how???

L273-275: "Asari by default does not return peaks of bad shapes, the 504 red *(sic)* features were due to the padded retention time (...) in the recomputing of peak shape". What does any of this even mean? 1) what is the "padded retention time", 2) "red" seems to be some arbitrary cutoff not specified in the text or the graph. The text in 270-282 seems to suggest that the graphs are not good at showing what they should show.

L303-304: This method is imprecise because scan-to-scan time can vary e.g. based on additional data-dependent MS2 scans between the MS1 scans. I suggest to at least multiply by Δt (to implement the simplest version of numerical integration).

L378 "but a 2nd order model may be required to cover the full m/z range". What motivates this confusing statement?

L296 I have no idea what I'm supposed to learn from Figure 6C, since all points are so strongly overlaid on each other. On such dense data, point size and color of individual points are very hard to identify and to make sense of. I'm not sure there is a good way to visualize four dimensions at the same time for this; less is probably more (like in D, E, F). However, I'm also not sure how to interpret 6D, E, F from a user perspective and what to learn from them.

SI: Such a detailed description of the experiments and LC-MS! If only the algorithms (the actual topic of the paper) were also described as precisely!

Reviewer #2 (Remarks to the Author):

My concerns still stand. While this revised version now includes the comparison with other software, the manuscript practically does not contain any description of the methods/algorithms of Asari and the methods and description of how Asari and XCMS/mzMine were compared. It is surprising that the authors have not described the core algorithms, despite the lack of details being a concern raised by both reviewers. I believe that the authors have written this manuscript with too much salesmanship and without focusing on the methods or the intellectual part of the work.

In another of my previous concern I suggested: (...) "This would also enable performing correlations among adducts, in-source fragments and isotopes, to assess the quantitative performance."

The authors reply: "The correlations among adducts, in-source fragments and isotopes are an interesting idea, but there are also ongoing debates (e.g., Mahieu, et al. 2016.) (...) "Since quantification is not among the top issues discussed in this work, we prefer to keep it simple."

I also find surprising that the authors believe that the quantification of the peak area is not one of the main issues of this work. The aim of Asari is exactly that: detect peaks and enable their relative quantification among samples. Therefore, I disagree with the author's response.

With the dataset that authors have used (Nature methods 18.11 (2021): 1377-1385), the authors now have 314 correctly identified metabolites. These metabolites have isotopes, adducts and in-source fragments for which their relative concentration should correlate among samples, i.e., the M+H of a given metabolite and their M+Na or M+H-H₂O peaks should correlate among samples. How good is this correlation? Does Asari detect/misses more isotopes, adducts, in-source fragments than XCMS or mzMine? This would provide a more solid and practical comparison of the software with the other established tools. Of course, in some cases, the M+H and their adducts could not correlate well, but if Asari's quantification yields a higher correlation than XCMS or mzMine, it suggests that Asari performance is better.

Point-by-point response to reviewers' comments

We thank both reviewers for the constructive critiques and detailed discussions. They really helped improve the clarity, which is an inherent challenge in works of this breadth, as the communities are still working on the terminologies. Even the word “quantification” has different meanings to computational processing and to experimental metabolomics. The writing of this manuscript incorporated significant efforts to dissect and explain the complex issues for general readers. We hope to strike a good balance here between the perspectives from chemists and biomedical data scientists.

Reviewer #1 (Remarks to the Author):

****General:****

- 1) I appreciate the much more extensive writeup in the new version.
- 2) I believe I kind of understand what the algorithm is doing now. However I still think the authors are not describing the algorithm at the necessary level of detail.

A new section is added to the Supplementary Information to describe step-by-step details of the algorithms, Lines 264-454.

Full documentation is released on Readthedocs, including description of over 130 functions: <https://asari.readthedocs.io/>.

- 3) I have some issues with the writing. The authors should refer to e.g. XCMS or ADAP manuscripts for the precision and clarity expected.

Improvement of writing is described below.

- 4) With regards to the concept of "mSelectivity", I believe the authors are conceptually confusing a few things and need to clear things up a bit.

We disagree on the stated conceptual confusion of “mSelectivity”. The reasons are explained below under the specific comment.

In summary:

- I think the algorithm is great and deserves publication. (I also think the authors should acknowledge where potential weaknesses are and be realistic that it doesn't solve all problems in the world.)

Expanded in Discussion on potential weaknesses, Lines 391-397.

- However, the authors should bring this to the quality of a typical computational manuscript, get rid of buzzwords and hyperbole, and clear up the writing. It seems that the article is written from a quite "internal" perspective from a group that has worked on this for a while. It would benefit of the perspective of someone external, ideally someone who has experience with computational/algorithm manuscripts.

Per the suggestion by the reviewer, the revised version has also been reviewed by an external expert (the PI of MetaboAnalyst at McGill University) and a new data scientist in our group. A principal scientific writer at our institution also helped edit the manuscript.

As I understand it, the algorithm works as follows:

****MassGrid detection****

- 1) Mass track detection: For each sample, ad-hoc m/z bins (or clusters) found by the equivalent of single-linkage clustering with a fixed height on the m/z axis (L137-138)
- 2) Alignment of mass tracks across samples for m/z correction by an unclear method (L140-146)
- 3) "Mass tracks" to "Aligned mass tracks": Merging of (corrected) mass tracks across samples. It is not clear if this again uses the equivalent of single-linkage clustering.
- 4) Per-sample RT correction by lowess on selected features from aligned mass tracks (L164-L167)
- 5) "Aligned mass tracks" to "Composite mass tracks": RT-corrected aligned mass tracks are summed to a "composite mass track". It is unclear how one arrives at discrete time bins over which to sum intensities, or how this is otherwise done. These "composite mass tracks" together are then also called a "MassGrid". (L162-171)

This is now described in the new section in Supplementary Information, as step-by-step details.

Asari uses scan numbers as discrete time bins throughout, for computational efficiency. They are only converted to time (seconds) in reporting the result.

****Peak detection per composite mass track****

Elution peak detection seems to play a very important role, but is very poorly described.

- 6) There is some baseline and noise estimation (L254) It is unclear whether this uses the SNR definition from L290-292 or what it does. "The statistics then drive the decisions on baseline subtraction, detrending or smoothing", with no description of the decisions.
- 7) A mass track is then separated into segments of valid signals (L259-260), how?
- 8) Peak detection per segments with local maximum search, the rest remains vague

Finally, computation of peak shape, SNR and "cSelectivity", which serve for later inspection but don't have an influence on the processing itself.

This is now described in the new section in Supplementary Information, as step-by-step details.

It is correct that the three metrics are not part of peak detection but used as quality control afterwards.

I need to painfully piece together this information, and it remains quite incomplete. To me, it's quite clear: either an existing algorithm is used, then it should be referenced; otherwise, a new algorithm is used, and it should be described, e.g. in pseudocode. ****From the description, I should be able to write a bad implementation that does more or less the same thing****, some corner cases perhaps excluded but possibly mentioned. For an example of a decent and precise algorithm description, see e.g. the ADAP paper <https://doi.org/10.1021/acs.analchem.7b00947>, XCMS paper <https://doi.org/10.1186/1471-2105-9-504>, or the SI of <https://doi.org/10.3390/metabo6040039>.

We do appreciate the efforts made by the reviewers. With the step-by-step details in Supplementary Information and full documentation on Readthedocs, we hope the description is adequate.

We also appreciate that the reviewer brought up the examples of these papers, including the ADAP paper. Our lead author is also a coauthor on the ADAP papers. Although we don't take credit from the hard work by Dr. Xiuxia Du and her lab, being involved in the project, we can reflect on a couple of pertinent insights here.

First, the two back-to-back ADAP papers (Analytical Chemistry, 2017) were initially submitted as a single manuscript, but split in two because the editor thought it was too long. Our manuscript here covers a lot more ground than the ADAP papers, thus necessary to balance the level of details. In addition, our writing takes into consideration a broader readership.

Second, it is extremely difficult to compare different tools at lower levels, because a) many steps impact their results and they cannot be decoupled easily, and b) many parameters are involved and they are not defined or used in the same way. For example, as discussed below, a direct comparison of the wavelet algorithms in XCMS and MZmine is difficult because their input EICs are already different in the two tools. Therefore, the analysis in our manuscript is very careful to isolate the specific issues. For example, the peak shapes in Figure 5 had to be computed on the same set of EICs (mass tracks) and all features from all tools were verified to match to these EICs.

The parameter problem is also related to another discussion below.

Finally, **concerning the writing**: There is a lot of hyperbole, big statements and buzzwords in the paper. Further, some words are misused, like "cannot" instead of "should not" or a more idiomatic construct. The authors should revise this, less is more here.

We apologize for the impression of "hyperbole, big statements and buzzwords". Some languages are removed or justifications are spelled out. Complete agreement on "less is more here".

Overall, a major effort in this writing is to include the perspectives from biomedical data science. For people that are doing metabolomics data analysis, many are already familiar with a stack of data tools. It is critical to connect to these users and tools for the future growth of metabolomics. The stakeholders of metabolomics are many and diverse. A "buzzword" to one group could be just the daily reality to another group.

Just some examples. This is by far not exhaustive.

- "Asari provides verifiable elution peak detection". What does verifiable even mean here?

Changed to "Elution peak detection is verifiable and understandable in asari".

The information needed to verify the peak detection result can be looked up easily in the dashboard. More importantly, in the benchmark datasets, we were able to explain the reason for every feature that was missed by asari. From our experience, it is not easy to explain why features are missed in other tools. This predictable software behavior is the basis of being "verifiable".

- "A set of metrics fully describe data quality"

A user can adequately judge the peak quality based on these metrics. Our lab used to plot out all features of interest to inspect their quality – false positives were a major concern. With asari and these quality metrics, the burden of manual inspection is mostly removed. Explanation added to the text.

- "Asari delivers a new generation of computational performance" (aside from the hyperbole, what does this even mean?)

If any processing software in genomics achieves 10-100 times improvement of computational performance, nobody would dispute it as a new generation. Metabolomics studies are getting bigger and most human cohort studies require a sufficient sample size. The performance of asari has immediate impacts on infrastructure.

We have removed the words "new generation", however, not to make this a distraction.

- "Asari marks a clear transition"

Removed.

- "a bridge to data science", "bridge the gap to data science"

Besides the compatibility of computational tool stack, the people factor should be quite clear: there are many more skilled people in biomedical data science than in computational metabolomics. They can use asari out of box. But few of them would get over the learning curves of other metabolomics data processing tools, which are largely designed for chemists.

- "Computational processing cannot overstate the quality of analytical chemistry". What is this supposed to mean?

Rephrased to "important that data quality can be clearly assessed and that it is consistent with the quality of analytical chemistry".

Extant tools are prone to include bad features in the processing results, as illustrated in Figure 5B. These bad features often go into downstream data analysis and cause many problems.

- "Since asari considers all [...], confidence can be established at high level of sensitivity": What "confidence" are they talking about?

Removed. The "confidence" referred to probability of being true positive.

- "By linking features and raw data through composite map, asari produces an efficient solution of automated testing, debugging and verification". ??? What is tested automatically? What is debugged? What is verified? (Clearly the authors don't mean automated testing of the code, as the code repository doesn't seem to contain any unit tests.)

This refers to a design principle. In the ADAP papers, we reported two important bugs in previous software. But it took many dedicated months to identify one bug in XCMS and more months for one in MZmine. These were bugs that are unlikely to be caught in unit tests – computational experiments were carried out to track the altered output in large amount of data. With the explicit linking of objects in asari, these types of bugs should be trivial to identify.

****Regarding the mass track concept in general****

The key ideas of the paper are "mass tracks" and "mSelectivity". The notion is that

- 1) masses closer to each other than a preset resolution, described by mSelectivity, cannot reliably be distinguished
- 2) combining masses into mass tracks provides a reproducible basis for downstream analysis
- 3) peak detection on combined mass tracks instead of single samples is both more reproducible and more efficient.

While 2 and 3 are correct and great ideas, 1 confuses two things. Clearly, 528.2374 and 528.2399 (4.7 ppm apart) can easily and reproducibly be *distinguished* on the mass spectrometer (in two separate injections, or with different retention times), even if they might not be *resolved* in a single scan even on an Orbitrap. Peak picking with e.g. XCMS will reliably determine the two masses separately, which is important for formula assignment. Even TOFs with relatively low resolutions (20-30k) can determine the mass apex quite accurately.

I think it is a great idea to merge masses to mass tracks for processing. However, this should not be taken to mean that such masses are in fact all the same. The "mass track" method is, in a way, artificially recreating the analytical problem of "coalescence", which occurs for close masses in a single scan in Orbitraps (Gorshkov 2012 Rapid Commun Mass Spectrom., Kaufmann 2018 Rapid Commun Mass Spectrom.)

I would suggest, after peak detection, to go back and calculate the "real" m/z for a peak for the detected feature.

This is a great discussion, but we disagree on the stated conceptual confusion on "mSelectivity". The matter of debate is if a special case should be generalized.

The mSelectivity is not binary but a continuous score. If we assume a resolution of 5 ppm, the mSelectivity scores on 528.2374 and 528.2399, example given by the reviewer, are 0.6 not 0 using our formula. This is not a low score and suggests that they can be possibly distinguished. In Figure 1B, we were careful to use MZmine examples of 1 ppm, not 4.7 ppm.

It is true that some close peaks are well resolved on the instrument. But how do we know they are indeed different peaks? If they are distinguishable by LC retention time, the annotation step will treat them separately and in turn inform us the real m/z values. If they are not distinguishable by LC, the additional step of recalculating them, as suggested by the reviewer, will also risk of bringing many false positives from split peaks (not a rarity in centroiding).

The decision in asari is purely operational. At the step of constructing mass tracks, two peaks of 4.7 ppm apart could be combined into the same mass track. But at the end of the process, they will be found by a search using 5 ppm. As long as asari retrieves information in an expected way, it can be combined with other methods that use additional information. The more accurate m/z values can be obtained via identification and annotation efforts outside asari – it does not need to be in asari. If there is desire to resolve a subset of peaks, one can simply rerun asari using, e.g., 2 ppm.

We do not assume the ppm parameter applies to all peaks equally (related to another comment by this reviewer below). A preset ppm value will be wrong for some features, but the design of asari ensures that they will be found in the result within the same ppm used for processing.

We appreciate the interesting point on coalescence. The motivation of mass tracks is that there are finite and discrete mass values in a biological sample, since they have to obey the constraints of atomic combinations. The field is hopefully getting closer to this finite number with high-resolution instruments.

It's a great point that the masses are not always the same on a mass track. This is extended to the Discussion, Lines 401-411.

****Individual comments****

L120-123 There is no "manual" merging in XCMS as far as I am aware so that discussion is out of place

"XCMS has a step of merging neighbor peaks that are too close" refers to ``refineChromPeaks`` and ``MergeNeighboringPeaksParam`` in XCMS.

The paragraph refers "ad hoc" merging in general. It's a common practice not limited to XCMS. E.g., this sentence from the NetID paper (Chen et al, 2021) can be considered manual merging or filtering: "Redundant peak entries due to an imperfect peak-picking process are removed if two peaks are ≤ 0.1 min apart and their m/z difference is ≤ 2 ppm."

We changed "manual" to "ad hoc".

L140: "locks the origin of peaks in computational processes"

Changed to "serving as the parent level of peaks".

L141: Is the combination of sample mass tracks to composite mass tracks again equivalent to single linkage clustering, or is something else done there? I.e., if track A1 and A2 in sample A are 6 ppm apart, and track B3 in sample B is in between (after alignment), do they all get merged to a single composite mass track?

Tracks from the same sample, such as A1 and A2, are never merged during alignment in asari. Nearest neighbor clustering is used in asari, but small studies rely more on pairwise alignment that prioritize landmark tracks. Details are now added to Supplementary Information.

L145-146: "We can already use $^{13}\text{C}/^{12}\text{C}$..." Is this actually done, and if yes, how? Or is this a future possibility? The methods don't show this.

Yes, m/z alignment prioritizes $^{13}\text{C}/^{12}\text{C}$ patterns and Na/H patterns, i.e. landmark tracks. Details are now included in Supplementary Information.

L172: "Previous tools usually require the presence in multiple samples to call a feature, because of their high error rates": This is simply not true. Users are free to set whatever threshold they wish for feature

sample presence.

Removed.

L181ff: I tried the viewer. It seems that one can only look at composite mass tracks, not at individual samples? This would be helpful to understand if e.g multiple peaks could come from an RT alignment problem.

It is a design decision in asari not to include mass tracks from individual samples in the dashboard. Because it is not feasible for larger studies: very significant storage is required; to make them interactive, a large and responsive database would require, not a trivial cost. The composite mass tracks are adequate in most cases, helped by the consistent peak definition across samples. Should one wish to look at individual samples, example Jupyter notebooks are available in our code repository.

It is an important suggestion about the inspection of RT alignment. However, by looking at a group of peaks across samples, one would not have enough information to determine if there are actually two features or it is an artifact from alignment error. This usually requires information from known metabolites or spike-in internal standards. Nonetheless, we agree that specific information on RT alignment is important. Towards this end, we have added a new tab to the Dashboard to include an RT alignment plot in XCMS style (screen shot below, version 1.11.3).

We have also added an option to export the alignment of landmark peaks in each sample as a static figure, one figure per sample. This is turned on by the `--debug` flag in asari. An example is shown below. The outliers did not contribute to building the alignment function.

L187: "the computational complexity increases quickly [...] if all intermediate steps are recorded" This doesn't make sense, the computational complexity stays exactly the same. Maybe the memory complexity changes if stuff is kept in RAM, which was probably the case in MZMine 2 (and is quite unnecessary, I agree). WRT computational complexity, are the authors aware of things like big-O notation?

Changed to "demand of computational resources". While this does not affect the complexity of executing the core algorithm, it would necessitate either serious use of a database or disk caching for larger studies.

L204ff: The SI show that extremely different parameter sets were used for the different workflows, for example $1e5$ (!) minimal peak height for asari, $1e3/1e4$ threshold for ADAP/MZMine, $1e3$ noise filter for XCMS, $5e3$ for MS-DIAL. Is there any rationale for this? This makes the results from Fig4 look somewhat arbitrary. Given that there is an obvious sensitivity parameter, could one construct something like a ROC curve?

These parameters are rather consistent between MZmine/ADAP and XCMS, as the most critical are the same minimum peak height $1E4$ and the same m/z tolerance 5 ppm. The minimum intensity filter, called "noise" in XCMS CentWaveParam and "Group intensity threshold" in MZmine ADAP Chromatogram builder, is the same at $1E3$. This is the same as default in asari, "min_intensity_threshold" at $1E3$.

As discussed above, regarding our experience of participating in the ADAP papers, terminology and implementation varies between software tools. Some other parameters are not directly comparable. We could not find explicit definition of minimum intensity filter in MS-DIAL, therefore took a compromise on minimum peak height to use the midpoint of $1E3$ and $1E4$.

Asari has a default minimum peak height $1E5$, because it uses composite mass tracks, which have higher intensity than EICs in a single sample. It is a reasonable default value for Orbitrap data, as most projects have more than 10 samples. We did not lower it just to boost the result in this paper – as explained in Figure S5 that this higher default value led to one and only one fewer true feature detected in the Yeast2021 dataset (3 samples, not typical in real applications).

ROC curves are not possible because the true negatives are not known.

L238: "as so many confusions are related to peak detection as well as the coverage of metabolomics". What is this supposed to mean?

Rephrased to "because there are many confusions in the field and the coverage of metabolomics is related to how many peaks are detected". Now Lines 247-248.

L239-241: While ADAP uses wavelets too, the overall algorithm is obviously still quite different. The statement makes no sense. If two algorithms are the same, they will give the same result with the same parameters.

Elution peak detection is one of the many steps. As discussed above, the wavelet algorithms in XCMS and MZmine are not compared in isolation, because their input EICs are already different in the two tools. The results are sensitive to implementation details too.

L247: I think we can agree that Fig 5 top-right is a complex case; what does Asari detect there?

Screen shot of the mass track with seven detected peaks is added as Figure S8A.

L255: The statistics drive the decisions... how???

This is now described in the new section in Supplementary Information, as step-by-step details.

L273-275: "Asari by default does not return peaks of bad shapes, the 504 red *(sic)* features were due to the padded retention time (...) in the recomputing of peak shape". What does any of this even mean? 1) what is the "padded retention time", 2) "red" seems to be some arbitrary cutoff not specified in the text or the graph. The text in 270-282 seems to suggest that the graphs are not good at showing what they should show.

The "padded retention time" was described in the figure caption, now is also explained in the main text, Lines 284-286. In summary, peaks from all tools were mapped to the same set of mass tracks. Because the reported retention time differs slightly between tools, this calculation extended 3 seconds on each side of the peak range, which led to slightly worse fitting for some small peaks.

L303-304: This method is imprecise because scan-to-scan time can vary e.g. based on additional data-dependent MS2 scans between the MS1 scans. I suggest to at least multiply by Δt (to implement the simplest version of numerical integration).

The reviewer raised a special case. When MS2 scans interrupt MS1 scans, the data in our opinion are not suitable for ab initio detection of elution peaks. The peak shapes are distorted and become problematic to any software. The approach needs the justification from additional information that a real peak is known to exist in that specific region, which is not a regular LC-MS processing workflow.

The default sum method of peak area works well in regular cases (as in Figure 7), because the retention time range is determined after RT alignment in asari. Handling special cases is part of the continued software development. For now, we have added two more options to export peak area values, area

under smoothed curve and area from a fitted Gaussian model. They can be used via the `--peak_area` flag.

L378 "but a 2nd order model may be required to cover the full m/z range". What motivates this confusing statement?

The question of choosing delta m/z or ppm is a common one. The ADAP paper included three figures to suggest using delta m/z. But it was clear that neither is perfect. We note that the previous discussion in the ADAP paper was based on older instruments – actually the data were from our group. For newer instruments, ppm is more practical.

This is actually related to the above discussion on mass accuracy. The ppm resolution is not a constant across m/z range. This is easy to observe in data, and discussed in the literature, e.g. Cox, J. and Mann, M., 2009. Computational principles of determining and improving mass precision and accuracy for proteome measurements in an Orbitrap. *Journal of the American Society for Mass Spectrometry*, 20(8), pp.1477-1485 (Figure 5a).

Extended discussion is added to Lines 404-411.

L296 I have no idea what I'm supposed to learn from Figure 6C, since all points are so strongly overlaid on each other. On such dense data, point size and color of individual points are very hard to identify and to make sense of. I'm not sure there is a good way to visualize four dimensions at the same time for this; less is probably more (like in D, E, F). However, I'm also not sure how to interpret 6D, E, F from a user perspective and what to learn from them.

Every application project needs visual summaries of quality control. They are practical needs on a daily basis in a facility setting or for data analysts. We have added Figure S9 to contrast "good" vs "bad" data. With a few examples, users can tell easily if the data quality is comparable to other studies, and if excessive low-quality peaks are present.

SI: Such a detailed description of the experiments and LC-MS! If only the algorithms (the actual topic of the paper) were also described as precisely!

As mentioned above, a new section is added to the Supplementary Information to describe step-by-step details of the algorithms. Full documentation is released at <https://asari.readthedocs.io/>.

Reviewer #2 (Remarks to the Author):

My concerns still stand. While this revised version now includes the comparison with other software, the manuscript practically does not contain any description of the methods/algorithms of Asari and the methods and description of how Asari and XCMS/mzMine were compared. It is surprising that the authors have not described the core algorithms, despite the lack of details being a concern raised by both reviewers. I believe that the authors have written this manuscript with too much salesmanship and without focusing on the methods or the intellectual part of the work.

As stated above, a new section is now added to the Supplementary Information to describe step-by-step details of the algorithms. Full documentation is released on <https://asari.readthedocs.io/>, including description of over 130 functions.

As discussed above and previously, this article is written for a broad audience. It is necessary to define and discuss the issues that drive the software design. We believe that the investigation of the longstanding issue of reproducibility and how design decisions are made is the intellectual part of the work.

In another of my previous concern I suggested: (...) “This would also enable performing correlations among adducts, in-source fragments and isotopes, to assess the quantitative performance.”

The authors reply: “The correlations among adducts, in-source fragments and isotopes are an interesting idea, but there are also ongoing debates (e.g., Mahieu, et al. 2016.” (...) “Since quantification is not among the top issues discussed in this work, we prefer to keep it simple.”

I also find surprising that the authors believe that the quantification of the peak area is not one of the main issues of this work. The aim of Asari is exactly that: detect peaks and enable their relative quantification among samples. Therefore, I disagree with the author’s response.

With the dataset that authors have used (Nature methods 18.11 (2021): 1377-1385), the authors now have 314 correctly identified metabolites. These metabolites have isotopes, adducts and in-source fragments for which their relative concentration should correlate among samples, i.e., the M+H of a given metabolite and their M+Na or M+H-H₂O peaks should correlate among samples. How good is this correlation? Does Asari detect/misses more isotopes, adducts, in-source fragments than XCMS or mzMine? This would provide a more solid and practical comparison of the software with the other established tools. Of course, in some cases, the M+H and their adducts could not correlate well, but if Asari’s quantification yields a higher correlation than XCMS or mzMine, it suggests that Asari performance is better.

We do not dispute that quantification is important. Among the many steps in data processing, if other steps are equal, quantification is the simplest and should not be a top concern. After all, it is numerical integration of the peak area or approximation of that, therefore every tool should have similar performance. Peak quantification is not the problem that impedes the progress. The major problems are caused by false positives and false negatives, by software crashes on larger datasets.

Nonetheless, we performed the analysis asked by this reviewer, as shown in Figure S6. The levels of correlation between ¹²C/¹³C isotopes are similar between the tools, as expected, while those for the Na/H adducts are sparser. The difference is the number of features, in which asari is more than comparable to XCMS and MZmine.

REVIEWERS' COMMENTS

Reviewer #1 (Remarks to the Author):

The manuscript is much better now. The added details make it possible to understand what the algorithm is doing and the rewrites markedly improve the reading experience.

I have two remaining remarks:

* L44-46: The authors mention that other algorithms have “strong dependence on complex parameters”, which is true. However the newly added SI makes it clear that their own algorithm has a lot of complex parameters too (SI L362f) except that they are treated as a given – let’s see how well those hold up to real-world cases! A truly parameter-free workflow would not need “magic ratios” such as 10%, quartiles, preset ceilings etc that might work for one dataset but not for another. Same at L387, they can say that the approach is “statistically minded” but in reality there are just more magic numbers hidden in the background. The mass tracks concept is still a great advance without needing to claim that it is low-parameter.

* Wrt the mSelectivity concept:

"If they are not distinguishable by LC, the additional step of recalculating them, as suggested by the reviewer, will also risk of bringing many false positives from split peaks (not a rarity in centroiding)"

No, I suggest to only recalculate the m/z values of final features which will not result in any additional split peaks.

"If they are distinguishable by LC retention time, the annotation step will treat them separately and in turn inform us the real m/z values."

Well, this is *exactly* the problem – how will annotation give us the correct result (between two conflicting close candidate metabolites) if it gets the incorrect m/z value as the input, shifted by as much as 5 ppm?! The info is lost when going to the mass track, so annotation would need to go back to the raw data, and we know this is not going to happen. At least in the examples SI L131ff the

"real" mass is not listed. This is *exactly* why I suggested to recalculate the mass for the final features. Mass accuracy and mass resolution are two different things, and a mass may be highly accurate while not being resolved from another close mass.

I leave it up to the authors whether they take up these comments; in the end it is good that this algorithm gets out into the wild, and the community can be the true arbiter!

Point-by-point response to reviewers' comments

Reviewer #1 (Remarks to the Author):

The manuscript is much better now. The added details make it possible to understand what the algorithm is doing and the rewrites markedly improve the reading experience.

I have two remaining remarks:

* L44-46: The authors mention that other algorithms have “strong dependence on complex parameters”, which is true. However the newly added SI makes it clear that their own algorithm has a lot of complex parameters too (SI L362f) except that they are treated as a given – let’s see how well those hold up to real-world cases! A truly parameter-free workflow would not need “magic ratios” such as 10%, quartiles, preset ceilings etc that might work for one dataset but not for another. Same at L387, they can say that the approach is “statistically minded” but in reality there are just more magic numbers hidden in the background. The mass tracks concept is still a great advance without needing to claim that it is low-parameter.

The baseline and noise levels are statistical estimates on the data and drive algorithmic decisions. All statistical models have parameters but they are fundamentally different from the parameters in previously tools, which are based on signal processing and less adaptive to data.

Local statistical parameters are part of the design but not a concern to end users. E.g., the default ceiling value in elution peak detection is $1E8$, which is a value for normalization to ensure stability of the algorithm. If the value is changed to $1E7$, there is no perceivable impact on the results. It is local and does not propagate to the next step.

We clearly distinguished “tunable parameters” in the manuscript. The list of user tunable parameters had been explained in the user manual. We stated in the manuscript that “default parameters are designed to work with a broad range of data”. Power users can modify the parameters and workflows, but the majority of users are relieved of the burden of parameter tuning. Two sentences are added to L442 and L445 to clarify this.

* Wrt the mSelectivity concept:

"If they are not distinguishable by LC, the additional step of recalculating them, as suggested by the reviewer, will also risk of bringing many false positives from split peaks (not a rarity in centroiding)"

No, I suggest to only recalculate the m/z values of final features which will not result in any additional split peaks.

"If they are distinguishable by LC retention time, the annotation step will treat them separately and in turn inform us the real m/z values."

Well, this is *exactly* the problem – how will annotation give us the correct result (between two conflicting close candidate metabolites) if it gets the incorrect m/z value as the input, shifted by as much as 5 ppm?! The info is lost when going to the mass track, so annotation would need to go back to the raw data, and we know this is not going to happen. At least in the examples SI L131ff the “real” mass is

not listed. This is *exactly* why I suggested to recalculate the mass for the final features. Mass accuracy and mass resolution are two different things, and a mass may be highly accurate while not being resolved from another close mass.

I leave it up to the authors whether they take up these comments; in the end it is good that this algorithm gets out into the wild, and the community can be the true arbiter!

If the reviewer assumed that the m/z values from asari were used for inferring chemical formulas or structures, this is not the case. Asari is for batch processing. Structural elucidation requires separate and dedicated efforts.

In a common scenario of metabolomics, the libraries of authentic standards and MS/MS are built by separate experiments and often different software. The bottom line is that the true m/z values can be calculated for compounds in the libraries and databases. When they are matched to asari results, asari features get their true values via annotation. Since the matching is using the same preset ppm range, the correct result is guaranteed to be found. This is the case when LC-MS adequately distinguishes the compounds.

If we want to push the limit of the LC-MS technology, we are getting into special cases that are difficult to generalize in high-throughput experiments. The example given by the reviewer, “two conflicting close candidate metabolites” not distinguished by LC and barely by mass, is a special case. The split m/z peaks (not elution peaks) from centroiding can look just like this special case. Without the prior knowledge of the existence of two metabolites, one may not be able to tell them from artifacts. The suggestion of recalculating m/z does involve this above decision. We consider such cases more suitable for customized workflow than for the default asari workflow. For a tradeoff between sensitivity and true discovery rate, asari favors the latter because it has less impact on data analysis and sensitivity is only useful when supported by annotation.

This is a valuable discussion and we have included it in the updated software documentation. New text is added to L466-468 for clarification.